# Visibility-derived aerosol optical depth over global land from 1980 to 2021

Hongfei Hao[1], Kaicun Wang[2], Chuanfeng Zhao[3], Guocan Wu[1], Jing Li[3]

[1]Global Change and Earth System Science, Faculty of Geographical Science, Beijing Normal University, Beijing 100875, China

[2]Institute of Carbon Neutrality, Sino French Institute of Earth System Science, College Urban and Environmental Sciences, Peking University, Beijing 100871, China

[3]Institute of Carbon Neutrality, Department of Atmospheric and Oceanic Sciences, School of Physics, College Urban and Environmental Sciences, Peking University, Beijing 100871, China

*Corresponding Author: Kaicun Wang (kcwang@pku.edu.cn)*

## Abstract

Long-term and high spatial resolution aerosol optical depth (AOD) data are essential for climate change detection and attribution. Global ground-based AOD observations are sparsely distributed, and satellite AOD retrievals have a low temporal frequency, as well low accuracy before 2000 over land. In this study, AOD is derived from hourly visibility observations collected at more than 5000 meteorological stations over global land from 1980 to 2021. The AOD retrievals of the Moderate Resolution Imaging Spectroradiometer (MODIS) onboard the Aqua Earth observation satellite are used to train the machine learning model, and the ERA5 reanalysis boundary layer height is used to convert the surface visibility to AOD. Comparisons with independent datasets show that the predicted AOD has correlation coefficients of 0.55 with AERONET ground observations at daily time scale. The correlation coefficients are higher at monthly and annual scales, which are 0.61 for the monthly and 0.65 for the annual, respectively. The visibility-derived AOD at station scale is gridded into a 0.5° grid by ordinary kriging interpolation. The mean visibility-derived AOD over the global land (-60°N-85°N), the Northern Hemisphere, and the Southern Hemisphere are 0.161, 0.158, and 0.173, with a trend of -0.0026/10a, -0.0018/10a, and -0.0059/10a from 1980 to 2021. For the regional scale, the mean (trend) of AOD are 0.145 (-0.0041/10a), 0.139 (-0.0021/10a), 0.131 (-0.0009/10a), 0.153 (-0.0021/10a), 0.192 (-0.0100/10a), 0.275 (-0.0008/10a), 0.177 (-0.0096/10a), 0.127 (-0.0081/10a), 0.177 (-0.0003/10a), 0.222 (-0.0000/10a), 0.232 (0.0071/10a), and 0.255 (0.0096/10a) in Eastern Europe, Western Europe, Western North America, Eastern North America, Central South America, Western Africa, Southern Africa, Australia, Southeast Asia, Northeast Asia, Eastern China, and India. The visibility-derived AOD at station and grid scales over global land from 1980 to 2021 are available at National Tibetan Plateau / Third Pole Environment Data Center (https://doi.org/10.11888/Atmos.tpdc.300822) (Hao et al., 2023).

How to cite. Hao, H., Wang, K., C. Zhao, Wu, G., J. Li (2023). Visibility-derived aerosol optical depth over global land (1980-2021). National Tibetan Plateau / Third Pole Environment Data Center. https://doi.org/10.11888/Atmos.tpdc.300822.

# 1 Introduction

Atmospheric aerosols are composed of solid and liquid particles suspended in the atmosphere. Aerosol particles are directly emitted into the atmosphere or formed through gas-particle transformation (Calvo et al., 2013), with diverse shapes and sizes (Fan et al., 2021), optical properties, and components (Liao et al., 2015; Zhang et al., 2020; Li et al., 2022). Most atmospheric aerosols are concentrated in the troposphere, especially in the boundary layer (Liu et al., 2022), with a high concentration near emission sources (Kulmala et al., 2004) , and a small portion are distributed in the stratosphere. Atmospheric aerosols severely impact the atmospheric environment and human health. They deteriorate air quality, reduce visibility, and cause other environmental issues (Wang et al., 2012; Boers et al., 2015). They impair human health or other organisms' conditions by increasing cardiovascular and respiratory disease incidence and mortality rates (Chafe et al., 2014; Yang et al., 2022). The Global Burden of Disease shows that global exposure to ambient $PM_{2.5}$ (particulate matter suspended in air with an aerodynamic diameter of less than 2.5 micrometers) resulted in 0.37 million deaths and 9.9 million disability-adjusted life years (Chafe et al., 2014).

Aerosols are inextricably linked to climate change. Atmospheric aerosols alter the Earth's energy budget and then affect the climate (Li et al., 2022). They cool the surface and heat the atmosphere by scattering and absorbing solar radiation (Forster et al., 2007; Chen et al., 2022). Aerosols, such as black carbon and brown carbon, also absorb solar radiation (Bergstrom et al., 2007), heat the local atmosphere and suppress or invigorate convective activities (Ramanathan et al., 2001; Sun and Zhao, 2020). Aerosols also alter the optical properties and life span of clouds (Albrecht, 1989). Atmospheric aerosols strongly affect regional and global short-term and long-term climates through direct and indirect effects (Mcneill, 2017).

Tropospheric aerosols are considered as the second largest forcing factor for global climate change (Li et al., 2022), and they reduce the warming due to greenhouse gases by -0.5°C (Ipcc, 2021). However, aerosols are also regarded as the largest contributor to quantifying the uncertainty of present-day climate change (Ipcc, 2021). The uncertainties are caused by the deficiencies of the global descriptions of aerosol optical properties (such as scattering and absorption) and microphysical properties (such as size and component), and the impact on cloud and precipitation, further affecting the estimation of aerosol radiative forcing (Lee et al., 2016; Ipcc, 2021). Therefore, sufficient aerosol observations are crucial. In aerosol measurements, aerosol optical depth (AOD) is often used to describe its column properties, which represents the vertical integration of aerosol extinction coefficients. AOD is an important physical quantity for estimating the content, atmospheric pollution and climatology of aerosols (Zhang et al., 2020).

AOD data usually from ground-based and satellite-borne remote sensing observations. They have both advantages and disadvantages. Ground-based lidar observation is an active remote sensing technology. Lidar generally emits laser and receives backscattered signals to invert the extinction coefficient of aerosols at different heights (Klett, 1985). By using the depolarization ratio, the type of aerosol, such as fine particles or dust, can be distinguished (Bescond et al., 2013). The AOD within a certain height can be calculated by integrating the extinction coefficients; however, scattering signals are usually not received near the ground, leading to blind spots (Singh et al., 2019).

At present, there are many ground-based lidar worldwide and regional networks, which provides important support of vertical changes in aerosols, such as the NASA Micro-Pulse Lidar Network (MPLNET) in the early 1990s (Welton et al., 2002), the European Aerosol Research Lidar Network (EARLINET) since 2000 (Bösenberg and Matthias, 2003), the Latin American Lidar Network (LALINET) since 2013 (Guerrero-Rascado et al., 2016).

Ground-based remote sensing observations supply aerosol loading data (such as AOD), by measuring the attenuation of radiation from the top of the atmosphere to the surface (Holben et al., 1998). This type of observation mainly uses weather-resistant automatic sun and sky scanning spectral radiometers to retrieve optical and microphysical aerosol properties (Che et al., 2014). The Aerosol Robotic Network (AERONET) is a popular global network composed of NASA and multiple international partners that provides high-quality and high-frequency aerosol optical and microphysical properties under various geographical and environmental conditions (Holben et al., 1998; Dubovik et al., 2000). The AERONET observations are extensively used to validate satellite remote sensing observations and model simulations, as well as climatology study (Dubovik et al., 2002b). There are many regional networks of sun photometers, such as the Maritime Aerosol Network (MAN), which use a handheld sun photometer to collect data over the ocean and is merged into AERONET (Smirnov et al., 2009), the China Aerosol Robot Sun Photometer Network (CARSNET) (Che et al., 2009), the Canadian sub-network of AERONET (AEROCAN) (Bokoye et al., 2001), Aerosol characterization via Sun photometry: Australian Network (AeroSpan) (Mukkavilli et al., 2019), and the sky radiometer network (SKYNET) in Asia and Europe (Kim et al., 2004; Nakajima et al., 2020). Another very valuable global network is the NOAA/ESRL Federated Aerosol Network (FAN), which uses integrated nephelometers distinct from sun photometers, mainly located in remote areas, providing background aerosol properties over 30 sites (Andrews et al., 2019).

Satellite remote-sensing is a space-based method that can provide aerosol properties worldwide. With the development of satellite remote sensing technology since 1970s, aerosol distributions can be extracted with the advantage of sufficient real-time and global coverage from multiple satellite sensors (Kaufman and Boucher, 2002; Anderson et al., 2005). The Advanced Very High Resolution Radiometer (AVHRR) is the earliest sensor used for retrieving AOD over ocean (Nagaraja Rao et al., 1989). The Moderate Resolution Imaging Spectroradiometer (MODIS), on board the Terra (launched in 1999) and Aqua (launched in 2002) satellites is a popular sensor with 36 channels, which have been used for AOD retrieval over both ocean and land based on the Dark Target and the Deep Blue algorithms (Remer et al., 2005; Levy et al., 2013). The latest MODIS AOD data version is the Collection 6.1, which provides global AOD over 20 years (Wei et al., 2019). There are also many other satellite sensors that can be used to retrieve AOD, such as the Polarization and Directionality of the Earth's Reflectances (POLDER) during 1996-1997, 2003 and 2004-2013 (Deuzé et al., 2000), Sea-viewing Wide Field-of-view Sensor (SeaWIFS) during 1997-2007 (O'reilly et al., 1998), the Multi-angle Imaging Spectroradiometer (MISR) on Terra since 1999 (Diner et al., 1998). The Cloud-Aerosol Lidar with Orthogonal Polarization (CALIOP) has also derived aerosols in the vertical direction since 2006 (Winker et al., 2009).

These measurements provide important data for studying the global and regional spatiotemporal variabilities and climate effect of aerosols. However, ground-based remote sensing observations only provide aerosol properties with low spatial coverage. There were only about 150 ground

stations worldwide in 2002 and even fewer sites were available for climate analysis (Holben et al.,
1998; Chu et al., 2002), which limited aerosol climate research by spatial coverage (Bright and
Gueymard, 2019). Satellite remote sensing overcomes the limitations of spatial coverage. The
AVHRR has been used to retrieve AOD since 1980, but it is limited by a few channel number, low
spatial resolution, and insufficient validation through ground-based observations before 2000 (Hsu
et al., 2017). Many studies have only investigated the trends and distributions of aerosols after 2000
(Bösenberg and Matthias, 2003; Winker et al., 2013; Xia et al., 2016; Tian et al., 2023), because of
the lack of long-term and global cover AOD products, which is the bottleneck for aerosol climate
change detection and attributions.

To overcome these limitations and enrich aerosol data, alternative observation data could be utilized
to derive AOD. Atmospheric horizontal visibility is a suitable alternative (Wang et al., 2009; Zhang
et al., 2020), because it has the advantages of the long-term records with a large number of stations
worldwide.

Atmospheric visibility is a physical quantity that describes the transparency of the atmosphere
through manual and automatic observations, and the automatic observations of visibility usually
measure atmospheric extinction (scattering coefficient and transmissivity). Koschmieder (1924)
first proposed the relationship between the meteorological optical range and the total optical depth.
Elterman (1970) futher established a formula between AOD and visibility by assuming an
exponential decrease in aerosol concentration with altitude, considering the extinction of molecules
and ozone to analyze air pollution, which called the Elterman model. Qiu and Lin (2001) corrected
the Elterman model by considering the influence of water vapor and used two water vapor pressure
correction coefficients to retrieve AOD of 16 stations in China in 1990. Wang et al. (2009) analyzed
the trend of AOD using visibility-based retrivals from 1973 to 2007 over land. Lin et al. (2014)
retrieved the AOD in eastern China in 2006 using visibility and aerosol vertical profiles provided
by GEOS-Chem. Wu et al. (2014) and Zhang et al. (2017) parameterized the constants in the
Elterman model and use satellite retrieved AOD to solve the parameters in the models at different
stations, to retrive the long-term AOD in China.

Zhang et al. (2020) reviewed the methods of visibility retrieval of AOD, indicating that visibility-
based retrieval of AOD can compensate for the shortcomings of long-term aerosol observation data.
Simultaneously, various parameters, such as station altitude, consistency of visibility data, water
vapor and aerosol vertical profiles (scale height), were discussed with modified suggestions
proposed. These studies have enriched AOD data regionally. These studies have enriched aerosol
data insome extent. At present, there are very few studies on global visibility-retrieved AOD and to
analyze climatology of aerosols.

The two physical quantities of visibility and AOD have both connections and differences, making it
challenging to retrieve AOD from visibility. Visibility represents the maximum horizontal visible
distance near the surface, while AOD represents the total vertical attenuation of solar radiation by
aerosols. The visibility of automatic observation is dependent on the local horizontal atmosphereic
extinction (Noaa et al., 1998). Visibility has not a simple linear relationship with meteorological
factors. The vertical structure of aerosols is the greatest challenge to obtain, as it is not a simple
hypothetical curve in complex terrain and circulation conditions (Zhang et al., 2020). These
limitations make it more complex to derive AOD. Machine learning methods can effectively address

complex nonlinear relationships between variables and have been widely applied in remote sensing and climate research fields. Li et al. (2021) used the random forest method to predict $PM_{2.5}$ in Iraq and Kuwait based on satellite AOD during 2001-2018. Kang et al. (2022) applied LightGBM and random forest to estimate AOD over East Asia, and the results showed a consistency with AERONET. Dong et al. (2023) derived aerosol single scattering albedo from visibility and satellite AOD over 1000 global stations. Hu et al. (2019) used a deep learning method to retrieve horizontal visibility from MODIS AOD. These studies have confirmed the ability of machine learning to effectively solve complex relationships among variables. And previous studies are mostly conducted at the regional or national scale, and few studies at the global scale. Thus, it is feasible to derive AOD from atmospheric visibility over global land by using the machine learning method.

In this study, we propose a machine learning method to derive AOD, where satellite AOD is the target value, and visibility and other related meteorological variables are the predictors. We explain the robustness of the model, validate the model's predictions using independent ground-based AOD, satellite retrievals and reanalysis AOD, and analyze the mean and trend of AOD across land and regions. Two datasets of long-term high-resolution AOD are generated. The Section 2 introduces the data and method. The Section 3 is the evaluation and validation of the visibility-derived AOD, and the distribution and trends are discussed at global and regional scales. The Section 5 presents the conclusions. This study is dedicated to supporting the research of aerosols in climate change detection and attribution.

## 2 Data and method

### 2.1 Study area

The study area is global land. A total of 5032 meteorological stations and 395 AERONET sites are selected in this study, shown in Figure 1. Twelve regions are selected for special analysis, including Eastern Europe, Western Europe, Western North America, Eastern North America, Central South America, Western Africa, Southern Africa, Australia, Southeast Asia, Northeast Asia, Eastern China, and India. The time range of the study is from 1980 to 2021, during which the records of meteorological stations are sufficient with a uniform spatial distribution. As shown in Figure 1, the daily records have exceeded 1500 stations, and monthly and annual records have exceeded 2000 during 1980-1990. After 2000, monthly records have reached 3000, which is the foundation of gridding AOD.

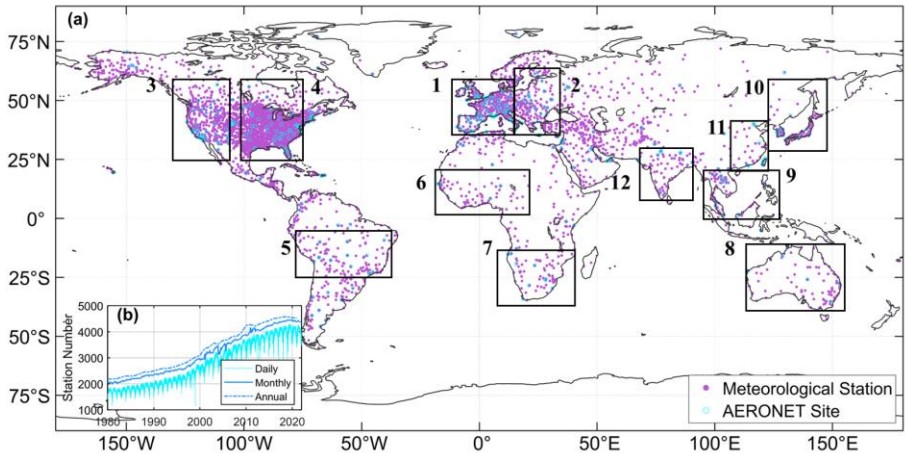

**Figure 1:** Study area (a) and the meteorological station number (b) with daily, monthly, and annual records. The number of meteorological stations (filled circles) is 5032. The number of AERONET sites (empty circles) is 395. The box regions of labelled with number 1-12 are Eastern Europe, Western Europe, Western North America, Eastern North America, Central South America, Western Africa, Southern Africa, Australia, Southeast Asia, Northeast Asia, Eastern China, and India.

**2.2 Meteorological data**

The ground hourly data from 1980 to 2021 is collected from 5032 automated meteorological stations of airports over land. Automated surface observations reduce errors associated with human involvement in data collection, processing, and transmission. The data can be downloaded at https://mesonet.agron.iastate.edu/ASOS. The data is extracted from the Meteorological Terminal Aviation Routine Weather Report (METAR). The World Meteorological Organization (WMO) sets guidelines for METAR reports, including report format, encoding, observation instruments and methods used, data accuracy, and consistency. These requirements ensure consistency and comparability of METAR reports globally. International regulations can be referenced at https://community.wmo.int/en/implementation-areas-aeronautical-meteorology-programme.
Among them, over 1,000 stations belong to the Automated Surface Observing System (ASOS), and others are sourced from airport reports around the world.

The daily average visibility is calculated using harmonic mean. Experiments have found that harmonic average visibility can better detect the weather phenomena than arithmetic average visibility (Noaa et al., 1998). The visibility is calculated using the extinction coefficient, which is directly proportional to the reciprocal of visibility (Wang et al., 2009). Harmonious average visibility can capture the process of visibility decline more quickly. Therefore, daily visibility will have greater representativeness:

$$V = n/(\frac{1}{V_1} + \frac{1}{V_2} + \cdots + \frac{1}{V_n}), \tag{1}$$

where V is the harmonic mean visibility, n = 24 for the daily visibility, and $V_1$, $V_2$,... $V_n$ are the individual hourly visibility.

In addition to hourly visibility (VIS), other variables closely related to aerosol properties are selected,

including relative humidity (RH), dew point temperature (DT), temperature (TMP), wind speed (WS) and sea-level pressure (SLP). Temperature affects atmospheric stability and the rate of secondary particle formation, and humidity influences the size and hygroscopic growth, and wind speed and pressure significantly impact the transport and deposition. Sky conditions (cloud amount) and hourly precipitation are also selected to remove the records of extensive cloud cover and precipitation.

We processed the data as follows. The records with high missing value ratio are eliminated (Husar et al., 2000). When over 80% overcast or fog, the records of sky conditions are eliminated, though such situations occur less than 1% of the time over land (Remer et al., 2008). The records with 1-hour precipitation greater than 0.1 mm are eliminated. We calculate the temperature dew point difference (dT). The low visibility records under "blowing snow" weather are eliminated at high latitude region (> 65°N), when wind speed is great than 4.5m/s (Husar et al., 2000). When the RH is greater than 90%, it is impossible to distinguish whether it is fog or haze, or both, and even precipitation. The records with RH greater than or equal to 90% are eliminated. When the RH is less than 30%, the dilution effect of aerosols is very low or even negligible. When RH is between 30% and 90%, visibility is converted to dry visibility (Yang et al., 2021c):

$$VISD = VIS/(0.26 + 0.4285 * log(100 - RH)), \qquad (2)$$

where VISD is the dry visibility.

Daily average of variables is calculated by at least 3 hourly records.

**2.3 Boundary layer height**

The hourly boundary layer height (BLH) from 1980 to 2021 is available from the Fifth Generation reanalysis of the European Medium-Range Weather Forecast Center (ERA5) with a resolution of 0.25° x 0.25° (https://cds.climate.copernicus.eu), which is the successor of ERA-Interim and has undergone various improvements (Hersbach et al., 2020). The atmospheric boundary layer is the layer closest to the Earth's surface and exhibits complex turbulence activities, and its height undergoes significant diurnal variation. The effects of the boundary layer on aerosols are mainly manifested in vertical distribution, concentration changes, transport, and deposition (Ackerman et al., 1995). The characteristics and variations in the boundary layer play a crucial role in regulating and adjusting the distribution of atmospheric aerosols. The boundary layer height serves as an approximate measure of the scale height for aerosols (Zhang et al., 2020).

Compared to observations of 300 stations over world from 2012 to 2019, the BLH of ERA5 was underestimated by 131.96m. Compared with the underestimated MERRA-2 (166.35m), JRA-55 (351.49m), and NECP-2 (420.86m), the BLH of ERA5 was closest to the observations (Guo et al., 2021). The BLH hourly data is temporally and spatially matched with the meteorological data before calculating the daily average.

Because the inverse of visibility is proportional to the extinction coefficient and positively related to AOD (Wang et al., 2009), we calculated the reciprocal of visibility (VISI) and the reciprocal of dry visibility (VISDI). Due to the influence of boundary layer height on the vertical distribution of particles (Zhang et al., 2020), we calculated the product (VISDIB) of the reciprocal of dry visibility and BLH. Therefore, the Predictor (Figure 2) is composed of 11 variables (TMP, Td, dT, RH, SLP, WS, VIS, BLH, VISI, VISDI, and VISDIB).

**2.4 MODIS AOD products**

Satellite daily AOD is available from the Moderate Resolution Imaging Spectroradiometer (MODIS) Level 3 Collection 6.1 AOD products of the Aqua (MYD09CMA) satellite from 2002 to 2021 and Terra (MOD09CMA) satellite from 2000 to 2021 with a spatial resolution of 0.05° x 0.05° at a wavelength of 550 nm (https://ladsweb.modaps.eosdis.nasa.gov). MOD/MYD09 has a higher spatial resolution than MOD/MYD08 (1° x 1°), which may result in a greater difference in AOD values and reduce the proximity ratio to match the visibility-derived AOD at station scale. Terra (passing approximately 10:30 am local time) and Aqua (passing approximately 1:30 pm local time) were successfully launched in December 1999 and May 2002, respectively.

MODIS, carried on the Terra and Aqua satellites is a crucial instrument in the NASA Earth Observing System program, which is designed to observe global biophysical processes (Salomonson et al., 1987). The 2,330 km-wide swath of the orbit scan can cover the entire globe every one to two days. MODIS has 36 channels and more spectral channels than previous satellite sensors (such as AVHRR). The spectral range from 0.41 to 15μm representing three spatial resolutions: 250 m (2 channels), 500 m (5 channels), and 1 km (29 channels). The aerosol retrieval algorithms use seven of these channels (0.47–2.13μm) to retrieve aerosol characteristics and uses additional wavelengths in other parts of the spectrum to identify clouds and river sediments. Therefore, it has the ability to characterize the spatial and temporal characteristics of the global aerosol field.

The MODIS aerosol product actually takes use of different algorithms for deriving aerosols over land and ocean. The Dark Target (DT) algorithm is applied to densely vegetated areas because the surface reflectance over dark-target areas was lower in the visible channels and had nearly fixed ratios with the surface reflectance in the shortwave and infrared channels (Levy et al., 2007; Levy et al., 2013). The Deep Blue (DB) algorithm was originally applied to bright land surfaces (such as deserts), and later extended to cover all cloud-free and snow-free land surfaces (Hsu et al., 2006; Hsu et al., 2013). MODIS Collection 6.1 aerosol product was released in 2017, incorporating significant improvements in radiometric calibration and aerosol retrieval algorithms.

The expected errors are $\pm (0.05 \pm 15\%)$ for the DT retrievals over land. Higher spatial coverage is observed in August and September, reaching 86-88%. During December and January, due to the presence of permanent ice and snow cover in high-latitude regions of the Northern Hemisphere, the spatial coverage is 78-80%. Thus, challenges remain in retrieving AOD values in high-latitude regions (Wei et al., 2019). However, visibility observations are available in high-latitude regions, thereby partially addressing the lack in these regions. In this study, the Terra and Aqua MODIS AOD are temporally and spatially matched with the meteorological stations. Aqua MODIS AOD is used as the Target, when training the model, and Terra MODIS AOD is used in the evaluation and validation of the model results, as shown in the flowchart (Figure 2).

**2.5 Ground-based AOD**

Ground-based 15-minute AOD data are available from the Aerosol Robotic Network (AERONET) Version 3.0 Level 2.0 product at 395 sites (Figure 1), which can be downloaded from https://aeronet.gsfc.nasa.gov. The AERONET program is a federation of ground-based remote sensing aerosol networks established by NASA and PHOTONS, including many subnetworks (such as AeroSpan, AEROCAN, NEON, and CARSNET). The sun photometer (CE-318) measures

spectral sun and sky irradiance in the 340-1020 nm spectral range. When the aerosol loading is low, the error is significant. When the AOD at 440 nm wavelength is less than 0.2, the error is 0.01, which is equivalent to the error of the absorption band in the total optical depth (Dubovik et al., 2002a). The total uncertainty in AOD under cloud-free conditions is less than ±0.01 for wavelength more than 440 nm, and ±0.02 for wavelength less than 440 nm (Holben et al., 1998). AERONET has three levels of AOD products: Level 1.0 (unscreened), Level 1.5 (cloud screened), and Level 2.0 (cloud screened and quality assured). Compared to Version 2, the Version 3 Level 2.0 database has undergone further cloud screening and quality assurance, which is generated based on Level 1.5 data with pre- and post-calibration and temperature adjustment and is recommended for formal scientific research (Giles et al., 2019). AERONET provides AOD products at wavelengths of 440, 675, 870, and 1020 nm. The AOD at 440nm and the Ångström index at 440-675nm are used for AOD at 550 nm not provided by AERONET, as shown in Eq. (3). AERONET AOD, as the 'true' value, is the average of at least two times within 1 hour (± 30 minutes) of Aqua transit time (Wei et al., 2019):

$$\tau_{550} = \tau_{440} \left(\frac{550}{440}\right)^{-\alpha}, \tag{3}$$

where $\tau_{440}$ and $\tau_{550}$ are the AOD at a wavelength of 440nm and 550 nm, and $\alpha$ is the Ångström index.

The matching conditions between AERONET sites and meteorological stations are (1) a distance of less than 0.5 ° (2) at least three years of observation. Finally, a total of 395 pairs were matched.

**2.6 AOD reanalysis dataset**

The monthly AOD (550nm) dataset of Modern-Era Retrospective Analysis for Research and Applications version 2 (MERRA-2) from 1980 to 2021 is a NASA reanalysis of the modern satellite era produced by NASA's Global Modeling and Assimilation Office with a spatial resolution of 0.5×0.625° (Gelaro et al., 2017), available at https://disc.gsfc.nasa.gov. MERRA-2 AOD uses an analysis splitting technique to assimilate AOD at 550 nm. AOD observations are including (1) AOD retrievals from AVHRR (1979-2002) over global ocean, (2) AOD retrievals from MODIS on Terra (2000–present) and Aqua (2002–present) over global land and ocean, (3) AOD retrievals from MISR (2000–2014) over bright and desert surfaces, and (4) direct AOD measurements from the ground-based AERONET (1999–2014) (Gelaro et al., 2017). The monthly MERRA-2 AOD is used to evaluate the model's predictive ability before 2000 and after 2000.

**2.7 Decision tree regression**

**2.7.1 Feature selection**

Although a multidimensional dataset can provide as much potential information as possible for AOD, irrelevant and redundant variables can also introduce significant noise in the model and reduce the model's accuracy and stability (Kang et al., 2021; Dong et al., 2023). Therefore, the F-test is used to search for the optimal feature subset in the Predictor, aiming to eliminate irrelevant or redundant features and select truly relevant features, which helps to simplify the model's input and improve the model's prediction ability (Dhanya et al., 2020). The F-test is a statistical test that gives an f-score(=-log(p), p represents the degree to which the null hypothesis is not rejected) by

calculating the ratio of variances. In this study, we calculate the ratio of variance between the Predictors and Target, and the features are ranked based on higher values of the f-score. A greater value of f-score means that the distances between Predictors and Target are less and the relationship is closer, thus, the feature is more important. We set p=0.05. When the score is less than -log (0.05), the variable in the Predictors is not considered.

### 2.7.2 Data balance

When it is clear, the AOD value is small, the variability of AOD is small (AOD<0.5), and the data is concentrated near the mean value. When heavy pollution, the AOD value is large (AOD>0.5). Compared to clear sky, the AOD sequence will show "abnormal" large values with low frequency, which is the imbalance of AOD data. When dealing with imbalanced datasets, because of the tendency of machine learning algorithms to perform better on the majority class and overlook the minority class, the model can be underfit (Chuang and Huang, 2023). Data augmentation techniques are commonly employed to address the issue in imbalance data, which applies a series of transformations or expansions to generate new training data, thereby increasing the diversity and quantity of the training data.

The Adaptive Synthetic Sampling (ADASYN) is a data augmentation technique specifically designed to address data imbalance problem (He et al., 2008; Mitra et al., 2023). It is an extension of the Synthetic Minority Over-sampling Technique (SMOTE) algorithm (Fernández et al., 2018). The goal of ADASYN is to generate synthetic sample data for the minority class to increase its representation in the dataset. ADASYN, which adaptively adjusts the generation ratio of synthetic samples based on the density distribution of sample data, improves the dataset balance and enhances the performance of machine learning models in dealing with imbalanced data.

The processing of imbalanced data includes (1) AOD sequences are classified into three types based on percentile (0-1%, 2% -98%, 99%), (2) When the mean of the third type of AOD is greater than 5 times the standard bias of the second type, it is considered an imbalanced sequence. These data, with a total amount less than 5% of the sample, are imbalanced data, and (3) Then synthetic samples are generated with the upper limit 10% of the samples.

### 2.7.3 Decision tree regression model

The decision tree is a machine learning algorithm based on a tree-like structure used to solve classification and regression problems. We adopt the CART algorithm to construct a regression tree by analyzing the mapping relationship between object attributes (Predictors) and object values (Target). The internal nodes have binary tree structures with feature values of "yes" and "no". In addition, each leaf node represents a specific output for a feature space. The advantages of the regression tree include the ability to handle continuous features and the ease of understanding the generated tree structure (Teixeira, 2004; Berk, 2008). Before training the tree model, the variables (Input) are normalized to improve model performance, and after prediction, the results are obtained by denormalization. The 10-fold cross-validation method is employed to improve the generalization ability of the model (Browne, 2000).

The core problems of the regression tree need to be solved are to find the optimal split variable and optimal split point. The optimal split point of Predictors is determined by the minimum MSE, which in turn determines the optimal tree structure. We set $Y = [y_1, y_2, \ldots, y_N]$ as the Target. We set $X = [x_1, x_2, \ldots, x_N]$ as the Predictors, $x_i = (x_i^1, x_i^2, \ldots x_i^n)$, i = 1,2,3 ..., N, where n is the feature number, and N is the length of sample. We set a training dataset as $D = [(x_1, y_1), (x_2, y_2), \ldots, (x_N, y_N)]$.

A regression tree corresponds to a split in the feature space and the output values on the split domains.
Assuming that the input space has been divided into M domains $[R_1, R_2, ..., R_M]$ and there is a fixed
output value on each $R_M$ domain, the regression tree model can be represented as follows:

$$f(x) = \sum_{m=1}^{M} c_m I(x \in R_M), m = 1, 2, ..., M, \tag{4}$$

where I is the indicator function, Eq. (5):

$$I = \begin{cases} 1, x \in R_m \\ 0, x \notin R_m \end{cases}, \tag{5}$$

When the partition of the input space is determined, the square error can be used to represent the
prediction error of the regression tree for the training data, and the minimizing square error is used to
solve the optimal output value on each domain. The optimal value $(\widehat{c_m})$ on a domain is the mean of the
outputs corresponding to all input, namely:

$$\widehat{c_m} = ave(y_i | x_i \in R_m), \tag{6}$$

A heuristic method is used to split the feature space in CART. After each split, all values of all features
in the current set are examined individually, and the optimal one is selected as the split point based on
the principle of minimum sum of the square errors. The specific step is described as follows: for the
training dataset D, we recursively divide each region into two sub domains and calculate the output
values of each sub domain; then, construct a binary decision tree. For example, split variable is $x^j$ and
split point is s. Then, in the domain $R_1(j, s) = [x|x^j \leq s]$ and domain $R_2(j, s) = [x|x^j > s]$, we can
solve the loss function $L(j, s)$ to find the optimal $j$ and $s$.

$$L(j, s) = \sum_{x_i \in R_1(j,s)} (y_i - c_1)^2 + \sum_{x_i \in R_2(j,s)} (y_i - c_2)^2, \tag{7}$$

When $L(j, s)$ is the smallest, $x^j$ is the optimal split variable and $s$ is the optimal split point for the
$x^j$.

$$\min_{j,s} \left[ \min_{c_1} \sum_{x_i \in R_1(j,s)} (y_i - c_1)^2 + \min_{c_2} \sum_{x_i \in R_2(j,s)} (y_i - c_2)^2 \right], \tag{8}$$

We use the optimal split variable $x^j$ and the optimal split point $s$ to split the feature space and calculate
the corresponding output value.

$$\widehat{c_1} = ave(y_i | x_i \in R_1(j, s)), \quad \widehat{c_2} = ave(y_i | x_i \in R_2(j, s)), \tag{9}$$

We traverse all input variables to find the optimal split variable $x^j$, forming a pair $(j, s)$. Divide the
input space into two regions accordingly. Next, repeat the above process for each region until the stop
condition is met. The regression tree is generated.

Therefore, the regression tree model $f(x)$ can be represented as follows:

$$f(x) = \sum_{m=1}^{M} \widehat{c_m} I(x \in R_M), m = 1, 2, ..., M, \tag{10}$$

**2.8 Gridding method**

Kriging is a regression algorithm to model and predict (interpolate) random processes/fields based on the
covariance function, which is widely used in geo-statistics (Pebesma, 2004). Ordinary Kriging is the

earliest and most extensively studied form of Kriging. It is a linear estimation system applicable to any intrinsic stationary random field that satisfies the assumption of isotropy. The two key parameters of Ordinary Kriging are the semi-variogram function and the weight factors (Goovaerts, 2000). It has been widely applied in fields, such as climatology, environmental science, and agriculture (Lapen and Hayhoe, 2003; Chen et al., 2010), due to high accuracy, stability, and insensitivity to data shape and distribution. This study utilizes area-weighted ordinary kriging algorithm to estimate the unknown values of AOD at specific locations to generate gridded AOD. The longitude range is between -179.5° E and 180 °E, the latitude range is between -60 °N and 85 °N, and the spatial resolution is 0.5 °*0.5 °.

Kriging variance represents the spatial correlation between different points, which is calculated by the semi variogram function (Goovaerts, 2000). Kriging variance is used to assess the spatial uncertainty of interpolation results, indicating the difference between predicted and true values. A higher kriging variance indicates fewer neighboring points and greater uncertainty, while a lower variance implies less uncertainty. To quantify the uncertainty of interpolation results, we provide the width of the confidence interval under the 95% confidence level based on kriging variance (Van Der Veer et al., 2009).

**2.9 Evaluation metrics**

Evaluation metrics, including Root Mean Squared Error (RMSE), Mean Absolute Error (MAE) and Pearson Correlation Coefficient (R), are used to measure the performance and accuracy of the model and gridded results.

$$RMSE = \sqrt{\frac{1}{n}\sum_{i=1}^{n}(y_i - \hat{y}_i)^2}, \tag{11}$$

$$MAE = \frac{1}{n}\sum_{i=1}^{n}|y_i - \hat{y}_i|, \tag{12}$$

$$R = \frac{\sum_{i=1}^{n}(y_i-\bar{y})(\hat{y}_i-\bar{\hat{y}})}{sqrt(\sum_{i=1}^{n}(y_i-\bar{y})^2\sum_{i=1}^{n}(\hat{y}_i-\bar{\hat{y}})^2)}, \tag{13}$$

where $y_i$ and $\bar{y}$ are the predicted value and the average of the predicted values. $\hat{y}_i$ and $\bar{\hat{y}}$ are the target and the average of the target. $i = 1,2,\ldots,n$. $n$ is the length of sample.

The expected error (EE) is used to evaluate the AOD derived from visibility.

$$EE = \pm(0.05 + 0.15 * \tau_{true}), \tag{14}$$

where $\tau_{true}$ is the AOD at 550 nm from AERONET, satellite and reanalysis datasets.

The width of 95% confidence interval (CI) is calculated from the kriging variance ($s^2$) (Van Der Veer et al., 2009) :

$$95\% \text{ CI} = 1.96 * \sqrt{s^2}, \tag{15}$$

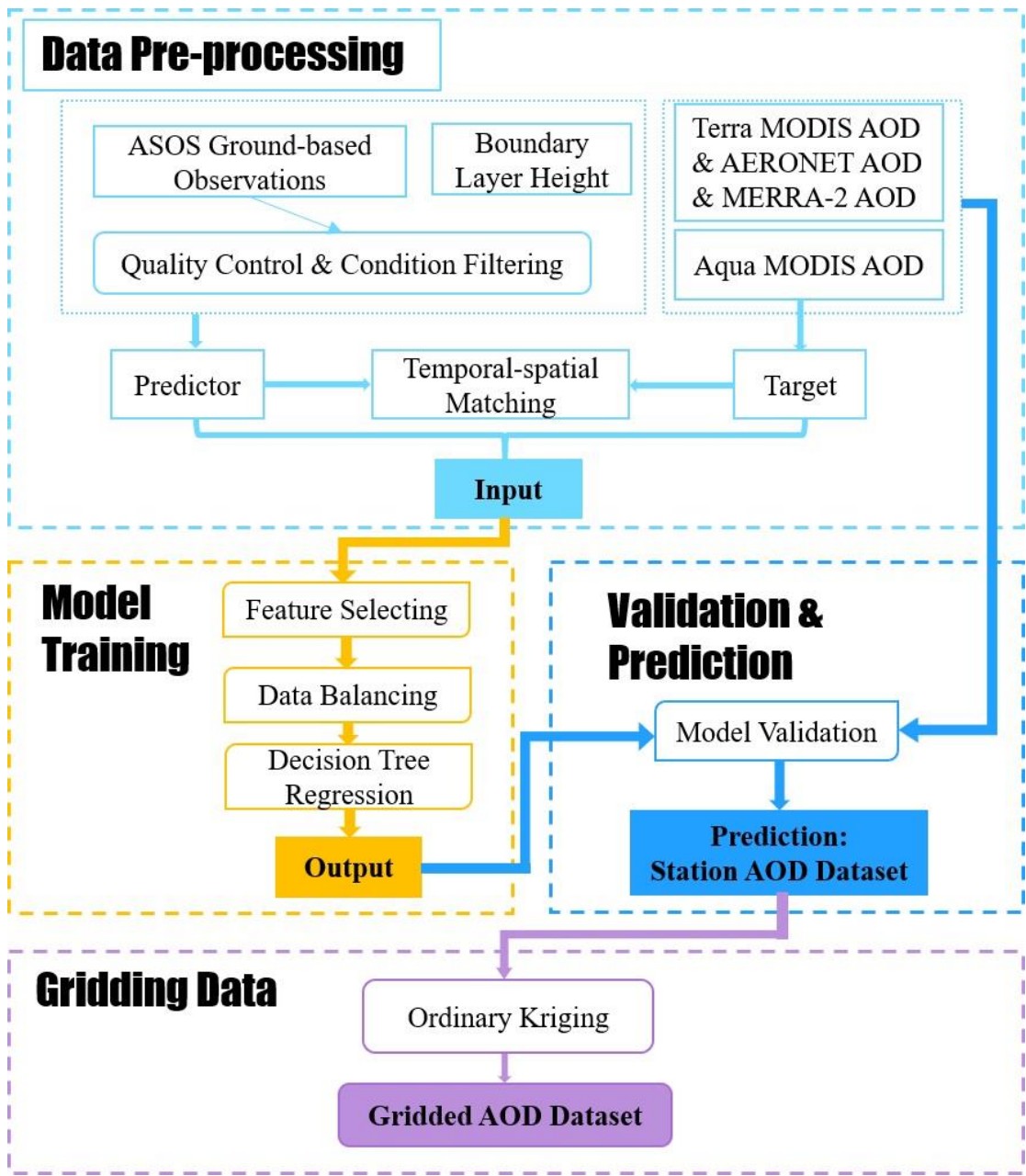

**Figure 2:** Flowchart for deriving aerosol optical depth (AOD).

**2.10 Workflow**

Figure 2 summarizes the flowchart and provides an overview of the structure of this study, which involves four main parts: (1) data preprocessing, (2) model training, (3) validation and prediction, and (4) data gridding.

# 3 Results and discussion

**3.1 Dependence of model performance on training data length**

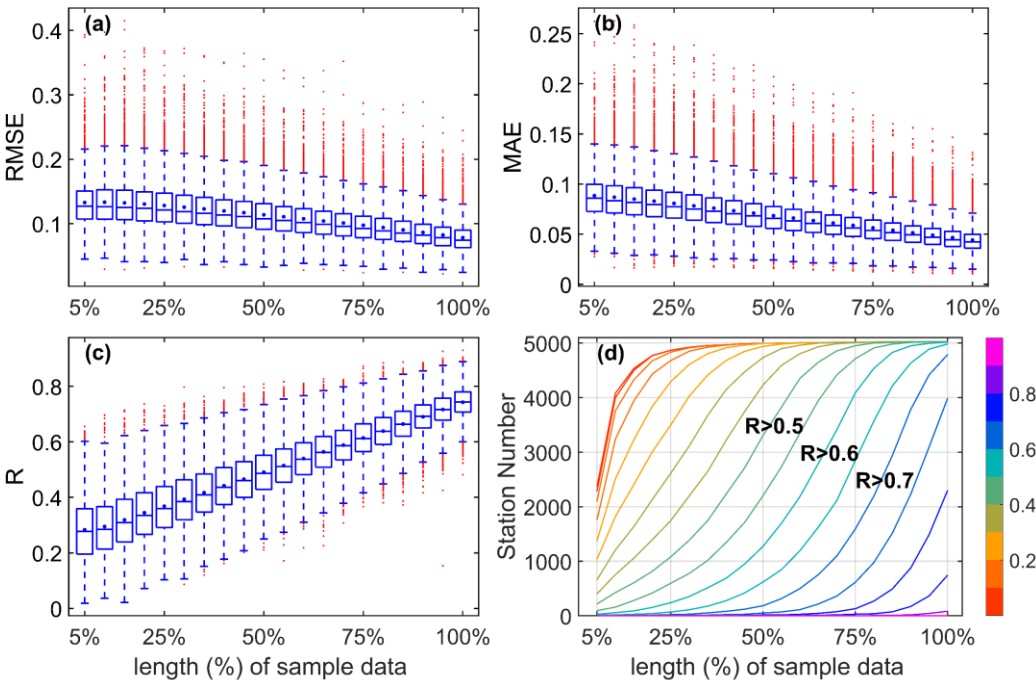

**Figure 3:** Boxplots of root mean squared error (RMSE) (a), mean absolute error (MAE) (b), and correlation coefficient (R) (c) between predicted values and target using different lengths of sample data (5% interval) as the training dataset, and the correlation coefficient curve (d) of the station number in the different lengths of sample data.

We build the models using different lengths of sample data (5% to 100%, with a 5% interval) by random allocation without overlap and evaluate the predictive performance of each model. Figure 3 depicts RMSE(a), MAE(b), and R (c) between the predicted values and target based on the training data of 5% to 100% sample data at a station. As the volume of the training data increases, the RMSE and MAE decrease, and the correlation coefficient increases. Compared to 5% of the sample data, the result of 100% sample data shows a decrease in RMSE by 41.1%, a decrease in MAE by 50.1%, and an increase in R by 162.3%. The relationship between the length of sample data and the model's performance is positive for each station. Figure 3 (d) shows that R of approximately 70% stations is greater than 0.5 at 50% of the sample data, while at 75%, the R of approximately 80% of stations is greater than 0.6. When 100% of the sample data is used as sample data, the R of approximately 80% of stations is greater than 0.75, and the R of about 97% is greater than 0.7. This finding indicates that the predictive capability and robustness of the model increase as the amount of training data increases. It may be attributed to the model's ability to capture more complex patterns and relationships among the input by multi-year data.

**3.2 Evaluation of model training**

Figure 4 shows the spatial distribution (a-c) and frequency and cumulative frequency (d-e) of RMSE, MAE, and R of all stations. The mean values of RMSE, MAE, and R are 0.078, 0.044, and 0.750, respectively. The RMSE of 93% stations is less than 0.11, the MAE of 91% is less than 0.06, and the R of 88% is greater than 0.7. The R values in Africa, Asia, Europe, North America, Oceania, and South America are 0.763, 0.758, 0.736, 0.750, 0.759, and 0.738, respectively. Although the RMSE and MAE of a few stations are high in America and Asia, the R is still high (>0.6). Therefore, the results of the

model's errors demonstrate that the model performs well on almost all stations.

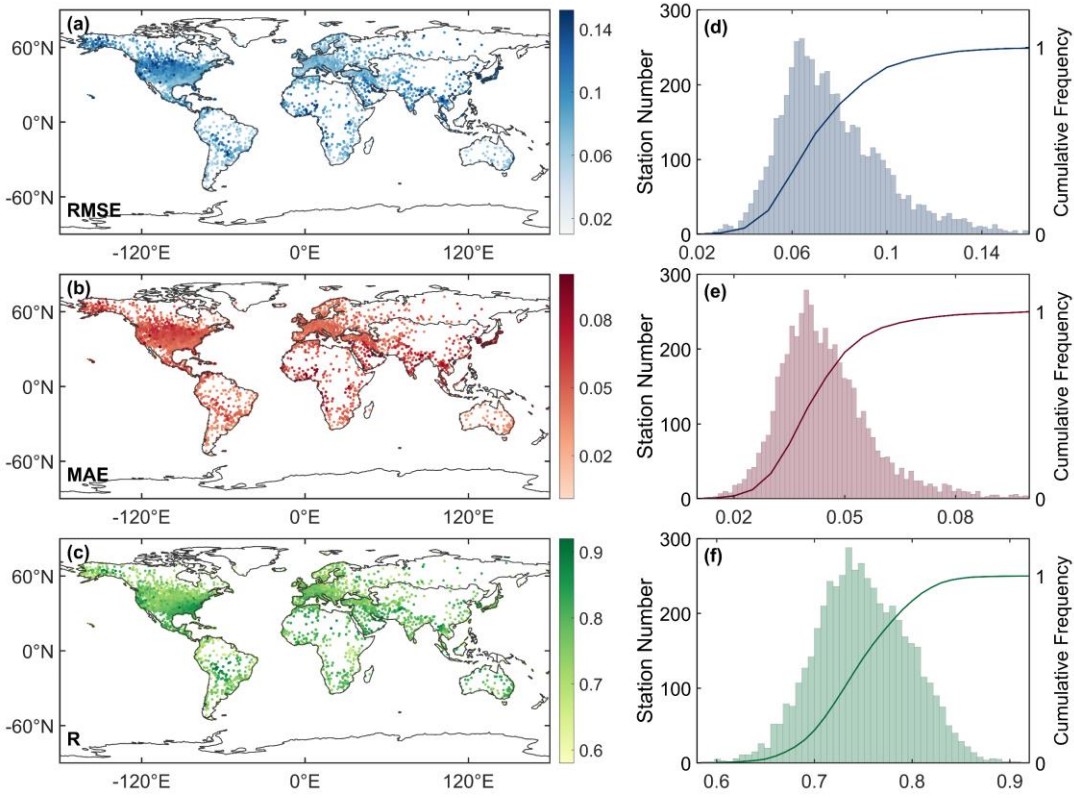

**Figure 4:** Spatial distribution (a-c) of root mean squared error (RMSE), mean absolute error (MAE), and correlation coefficient(R) between the model's result and target with 100% sample data. Station number (bar) and cumulative frequency (curve) (d-e) of RMSE, MAE, and R.

**3.3 Validation and comparison with MODIS and AERONET AOD**

**3.3.1 Validation over global land**

To validate the model's predictive ability, the visibility-derived AOD (for short, VIS_AOD) is compared with Aqua, Terra and AERONET AOD at 550nm for the global scale. Among them, Aqua AOD has been used as training data, which is not independent. Terra AOD and AERONET AOD have not been used as training data and can be regarded as independent data.

First, the relationship among daily MODIS and AERONET AOD is evaluated. Figure 5 shows the scatter density plots between AERONET AOD and Aqua AOD (a, d, g) and Terra AOD (b, e, h). The R values with Aqua AOD and Terra AOD are 0.643 and 0.637 on the daily scale, and 0.668 and 0.658 on the monthly scale, 0.658 and 0.665 on the yearly scale. The RMSE with Aqua AOD and Terra AOD are 0.158 and 0.163 on the daily scale, and 0.122 and 0.127 on the monthly scale, 0.101 and 0.103 on the yearly scale. The MAE values with Aqua AOD and Terra AOD are 0.084 and 0.088 on the daily scale, and 0.071 and 0.072 on the monthly scale, 0.061 and 0.062 on the yearly scale. The percentages of sample point falling within the EE envelopes are 64.66% and 62.54% on the daily scale, and 69.36% and 69.08% on the monthly scale, 74.80% and 75.89% on the yearly scale.

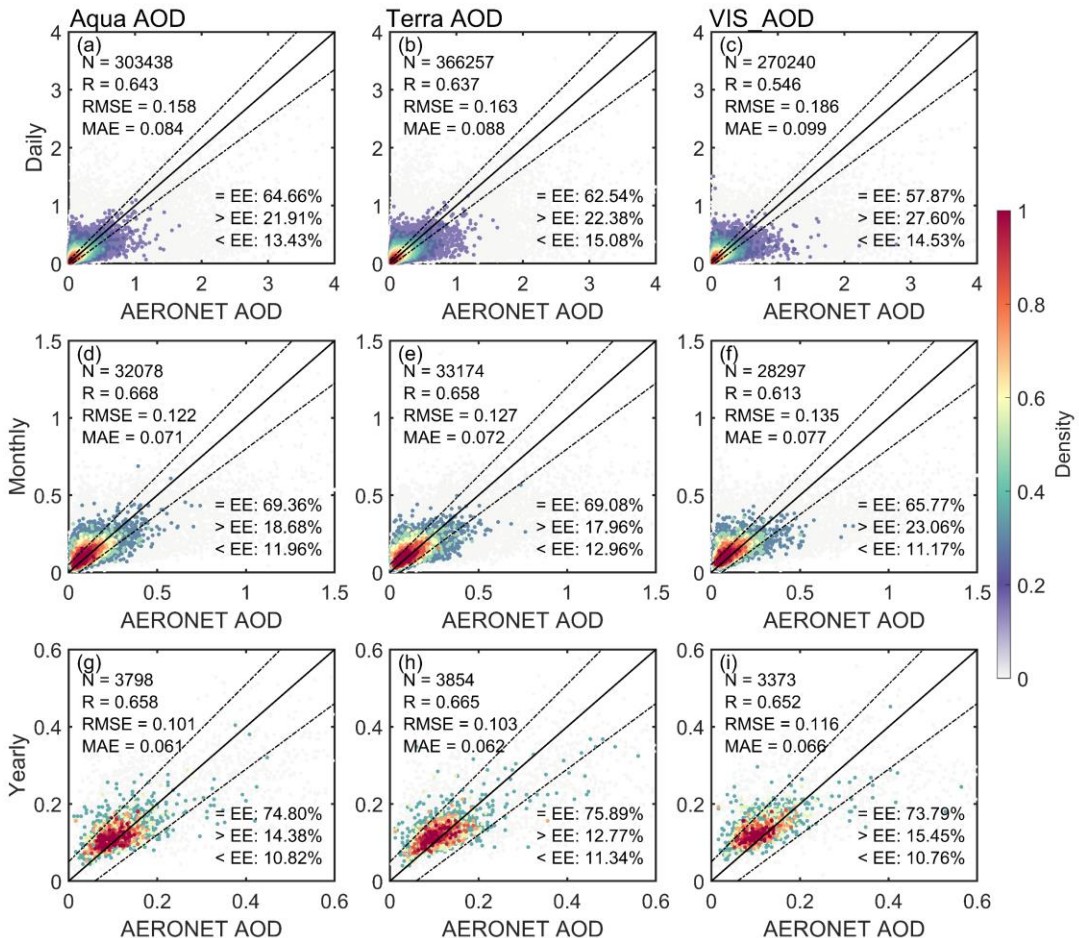

**Figure 5:** Scatter density plots between AERONET AOD (550nm) and Aqua MODIS AOD, Terra MODIS AOD and VIS_AOD on the daily (a-c), monthly (d-f) and yearly (g-i) scale. The solid black line represents the 1:1 line and the dashed lines represents expected error (EE) envelopes. The sample size (N), correlation coefficient (R), mean absolute error (MAE), and root mean square error (RMSE) are given. '= EE', '> EE', and '< EE' represent the percentages (%) of retrievals falling within, above, and below the EE, respectively. The matching time for Aqua AOD and VIS_AOD with AERONET AOD is 13.30 (± 30 minutes) at local time, and the matching time between Terra AOD and AERONET AOD is 10.30 (± 30 minutes) at local time.

Figure 6 shows the scatter density plots and the EEs between VIS_AOD and Aqua AOD, Terra AOD, and AERONET AOD. Aqua AOD is not an independent validation, and Terra and AERONET AOD are independent validation. For the daily scale, the R, RMSE and MAE of between VIS_AOD and Aqua AOD (15,962,757 pairs data) is 0.799, 0.079 and 0.044, respectively. The percentage of sample point falling within the EE envelopes is 84.12% on the global scale (Figure 6 a). The R between VIS_AOD and Terra AOD (17,145,578 pairs data) is 0.542, with a RMSE of 0.125 and MAE of 0.078. The percentage falling within the EE envelopes is 64.76% (Figure 6 b). The R between VIS_AOD and AERONET AOD (270,240 pairs data) at 395 sites is 0.546, with a RMSE of 0.186 and MAE of 0.099. The percentage falling within the EE envelopes is 57.87% (Figure 6 c).

For the monthly and annual scales, RMSE and MAE show a significant decrease between VIS_AOD and Aqua, Terra, and AERONET AOD, and R and percentages falling within EE show a significant increase

in Figure 6 (e-g, i-k). The monthly RMSEs are 0.029, 0.051, and 0.135, the monthly MAEs are 0.018, 0.031, and 0.077, and the R values are 0.936, 0.808, and 0.613, respectively. The percentages falling within the EE envelopes are 98.34%, 93.25%, and 65.77%. The RMSEs at the annual scale are 0.013, 0.024, and 0.116, the MAEs are 0.008, 0.015, and 0.066, and the R values are 0.976, 0.906, and 0.652, respectively. The percentages falling within the EE envelopes are 99.82%, 99.20%, and 73.79%. The percentage falling within the EE envelopes against AERONET is smaller than that against Terra, which may be related to the elevation of AERONET sites, the distance between AERONET and meteorological stations, and observed time. The results highlighted above demonstrate a clear improvement in performance on the monthly and annual scales compared to the daily scale (Schutgens et al., 2017), which provided a foundation for the gridded dataset.

To further examine the predictive capability of historical data, we compare the VIS_AOD with AERONET AOD before 2000, as shown in Figure 6 (d, h, l). We match 43 AERONET sites, with a total of 5166 daily records. The result indicates that the daily-scale R is close to that after 2000 (Figure 6 c), with the percentages approaching 50% falling within the EE envelopes. The monthly and annual correlation coefficients are even higher, with a percentage of 55% falling within the EE envelopes. Although the sample size is small, it still demonstrates the excellent predictive ability of the model. Compared with AERONET (an independent validation dataset), the performance of VIS_AOD is almost unchanged before and after 2000.

We also compare the VIS_AOD with the MERRA-2 reanalysis AOD on the monthly scales, as shown in Figure 7. The correlation coefficient between MERRA-2 and AERONET is 0.655 before 2000, slightly lower than the correlation coefficient (0.657) between VIS_AOD and AERONET. The correlation coefficient between MERRA-2 and AERONET is 0.829 after 2000, significantly higher than that before 2000, while the correlation coefficient between VIS_AOD and AERONET is 0.613. It suggests that VIS_AOD and MERRA-2 AOD have similar accuracy before 2000. The correlation of MERRA-2 after 2000 is higher and even performs better than MODIS retrievals (as shown in Figure 5) when evaluated at AERONET sites. However, before 2000, the correlation coefficient of MERRA-2 and AERONET, RMSE, and MAE all show significant changes and differences in consistency. The higher correlation between MERRA-2 and AERONET AOD is partly because MERRA-2 has assimilated AERONET AOD observations (Gelaro et al., 2017). Compared to AERONET, VIS_AOD and Aqua/Terra MODIS have a similar correlation coefficient. The correlation coefficient of VIS_AOD before 2000 is even higher than after 2000, and the changes in RMSE and MAE are not significant. It indicates good consistency of VIS_AOD. In conclusion, the predicted results have good consistency with AEONET AOD and Terra AOD on the daily scale. The monthly and annual results have a significant improvement. The model shows good predictive capabilities before/after 2000, highlighting the stable accuracy of VIS_AOD.

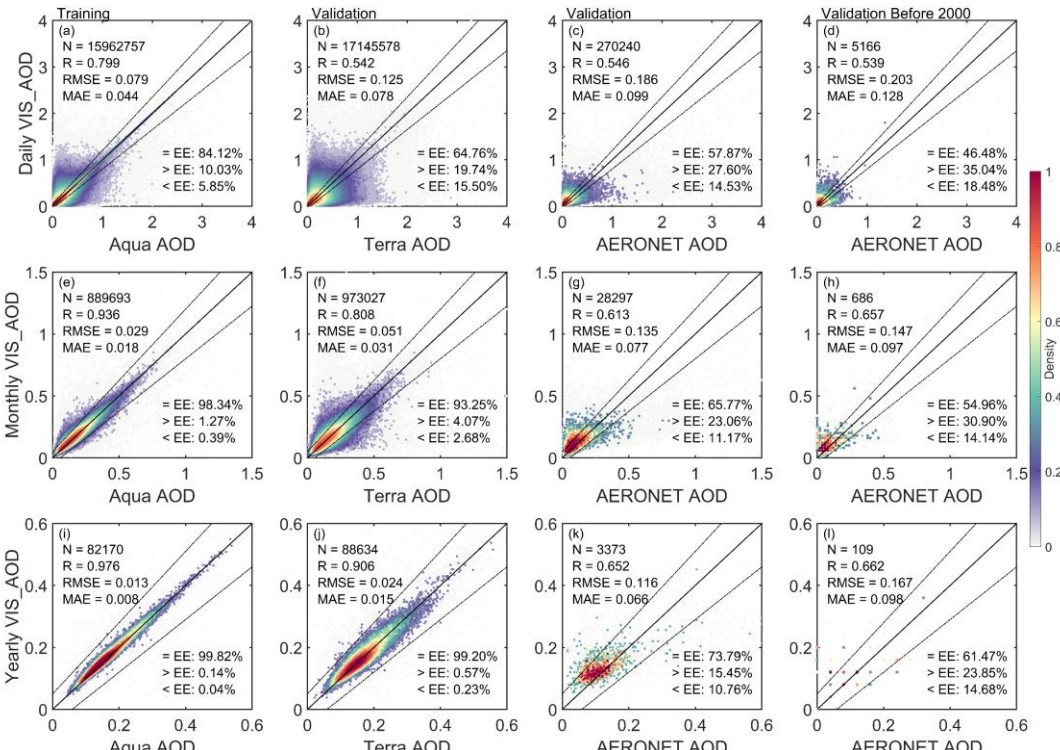

**Figure 6:** Scatter density plots between predicted AOD (VIS_AOD) and Aqua MODIS AOD, Terra MODIS AOD, AERONET AOD and AERONET AOD before 2000 on the daily (a-d), monthly (e-h) and yearly (g-i) scale. The solid black line represents the 1:1 line and the dashed lines represents expected error (EE) envelopes. The sample size (N), correlation coefficient (R), mean absolute error (MAE), and root mean square error (RMSE) are given. '= EE', '> EE', and '< EE' represent the percentages (%) of retrievals falling within, above, and below the EE, respectively. Note Aqua AOD is not an independent validation for predicted results, while Terra and AERONET are independent validation.

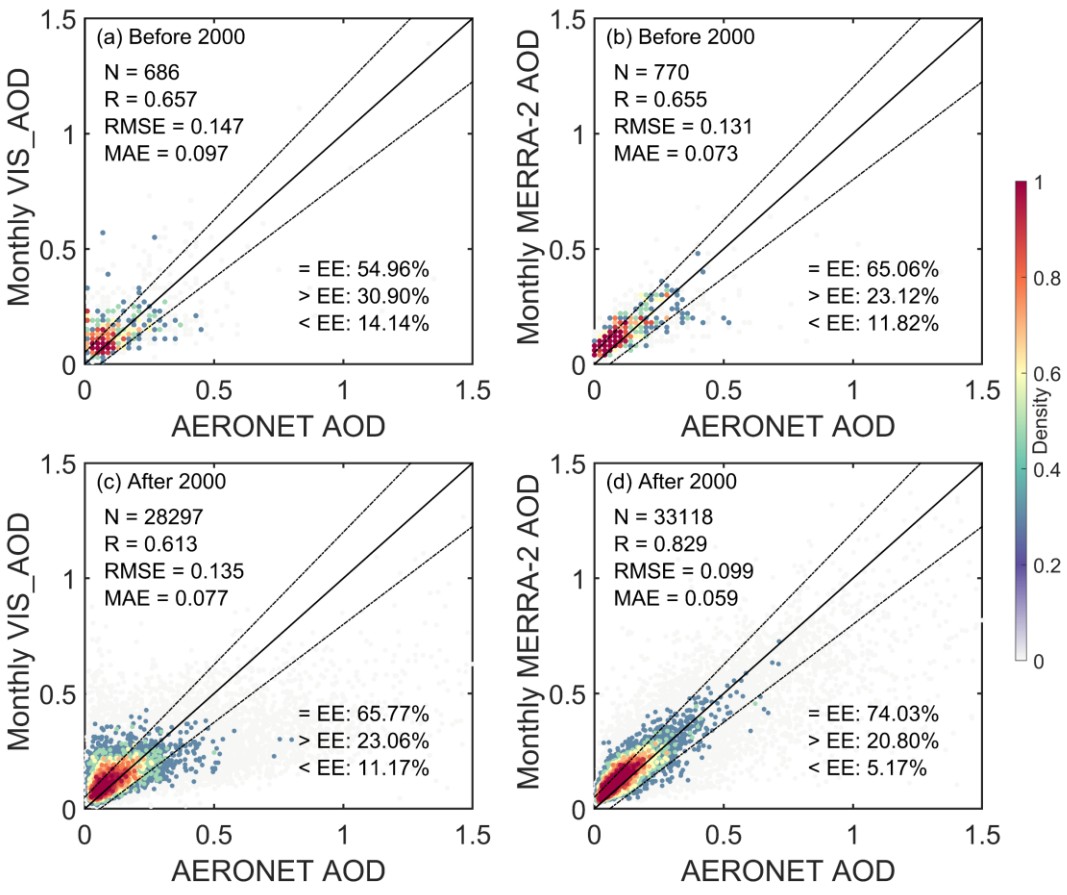

**Figure 7:** Scatter density plots between AERONET AOD and the predicted AOD (VIS_AOD) and
MERRA-2 AOD before/after 2000 on the monthly scale. The solid black line represents the 1:1 line and
the dashed lines represents expected error (EE) envelopes. The sample size (N), correlation coefficient
(R), mean absolute error (MAE), and root mean square error (RMSE) are given. '= EE', '> EE', and '<
EE' represent the percentages (%) of retrievals falling within, above, and below the EE, respectively.

### 3.3.2 Validation over regions

Aerosol loading exhibits spatial variability. Evaluation metrics for the relationships between
visibility-derived AOD and AERONET AOD and Terra AOD for each region are listed in Table 1.
Over Europe and North America, the results are similar to those of Terra and AERONET, with a
large number of data pairs, greater than $10^5$ (AERONET) and greater than $10^7$ except for Eastern
Europe (Terra) on the daily scale. Approximately 63% -70% fall within the EE envelopes. The
RMSE is approximately 0.1100, except for western North America, the MAE is approximately
0.0700, with a correlation coefficient between 0.44 and 0.54.
Over Central South America, South Africa, and Australia, data pairs are about $10^{3-4}$ (AERONET)
and $10^6$ (Terra) on the daily scale. 52-60% fall within the EE envelopes compared to AERONET,
and 58-67% compared to Terra. The RMSE is 0.03-0.05 compared to Terra, and 0.11-0.17 compared
to AERONET. The correlation coefficient ranges from 0.4 to 0.74, with the highest correlation
coefficient in South America at 0.740.
In Asia, India, and West Africa, the data pairs are only approximately $10^4$ (AERONET). 32% to 50%

fall within the EE envelopes compared to AERONET, the RMSE ranges from 0.2 to 0.5, and the

MAE ranges from 0.11 to 0.36. 51 to 58%, compared to Terra, fall within the EE envelopes, the

RMSE is around 0.16, and the MAE is around 0.11. Compared to AERONET, in these high aerosol

loading regions, RMSE and MAE increase, and the percentages falling within the EE envelopes

decrease, but the correlation coefficients do not significantly decrease.

Compared to Terra AOD, 55% -67% of data falls within the EE envelopes on the daily scale, 87% -

96% on the monthly scale, and over 97% on the yearly scale. Compared to AERONET AOD, 32-

68% of data falls within the EE envelopes, 24% -84% on the monthly scale, and 15% -97% on the

yearly scale. On both monthly and yearly scales, all metrics have shown a significant increase in

performance when compared to Terra. However, compared to AERONET, not all metrics increase

in some regions due to limited data pairs, such as West Africa, Northeast Asia, and India, which may

be due to the spatial differences between AERONET sites and meteorological stations.

### 3.3.3 Validation at a site scale

Sites, especially AERONET, are not completely uniform across the world or in any region, and

different stations have different sample sizes, which may lead to a certain uncertainty. Therefore,

further analysis was conducted on the spatial distribution of different evaluation metrics. Figure 8

shows the validation and comparison of daily VIS_AOD against Terra and AERONET AOD at a

site scale.

Compared to Terra daily AOD, the R of 67% stations is greater than 0.4, the mean bias of 83% is

**Table 1:** Evaluation metrics for the relationships between visibility-derived AOD and AERONET AOD and Terra AOD for each region.

| Region | | N | | | R | | | RMSE | | | MAE | | | Within EE (%) | | |
|---|---|---|---|---|---|---|---|---|---|---|---|---|---|---|---|---|
| | | daily | monthly | yearly | daily | monthly | yearly | daily | monthly | yearly | daily | monthly | yearly | daily | monthly | yearly |
| Eastern Europe | AERONET | 21724 | 2317 | 271 | 0.463 | 0.493 | 0.653 | 0.1069 | 0.0647 | 0.0326 | 0.0714 | 0.0442 | 0.0263 | 65.69 | 83.77 | 97.42 |
| | TERRA | 661630 | 36435 | 3278 | 0.464 | 0.665 | 0.790 | 0.1095 | 0.0471 | 0.0214 | 0.0726 | 0.0286 | 0.0122 | 66.07 | 94.71 | 99.18 |
| Western Europe | AERONET | 53043 | 6033 | 697 | 0.445 | 0.487 | 0.344 | 0.1089 | 0.0716 | 0.0513 | 0.0711 | 0.0474 | 0.0347 | 64.40 | 79.21 | 89.10 |
| | TERRA | 1778013 | 104620 | 9166 | 0.467 | 0.763 | 0.811 | 0.1096 | 0.0391 | 0.0210 | 0.0712 | 0.0268 | 0.0124 | 66.99 | 95.42 | 99.40 |
| Western North America | AERONET | 33859 | 2948 | 334 | 0.503 | 0.484 | 0.509 | 0.1465 | 0.0949 | 0.0566 | 0.0747 | 0.0597 | 0.0419 | 63.58 | 67.37 | 81.14 |
| | TERRA | 1725226 | 82734 | 7201 | 0.542 | 0.765 | 0.906 | 0.1144 | 0.0465 | 0.0180 | 0.0671 | 0.0267 | 0.0125 | 69.48 | 94.42 | 99.61 |
| Eastern North America | AERONET | 47407 | 5359 | 608 | 0.527 | 0.526 | 0.559 | 0.1135 | 0.0824 | 0.0436 | 0.0657 | 0.0472 | 0.0331 | 67.52 | 77.78 | 87.50 |
| | TERRA | 6280277 | 359520 | 31343 | 0.515 | 0.799 | 0.847 | 0.1159 | 0.0435 | 0.0165 | 0.0726 | 0.0275 | 0.0111 | 66.70 | 94.94 | 99.80 |
| Central South America | AERONET | 10911 | 1176 | 149 | 0.740 | 0.811 | 0.866 | 0.1735 | 0.1272 | 0.1060 | 0.1021 | 0.0904 | 0.0688 | 52.40 | 47.96 | 67.79 |
| | TERRA | 444780 | 26362 | 2410 | 0.545 | 0.820 | 0.776 | 0.1447 | 0.0591 | 0.0369 | 0.0909 | 0.0396 | 0.0219 | 58.48 | 89.29 | 97.39 |
| Southern Africa | AERONET | 4255 | 309 | 38 | 0.423 | 0.480 | 0.630 | 0.1553 | 0.1128 | 0.0705 | 0.1033 | 0.0805 | 0.0525 | 52.08 | 59.55 | 78.95 |
| | TERRA | 216239 | 11304 | 1118 | 0.518 | 0.821 | 0.870 | 0.1258 | 0.0511 | 0.0296 | 0.0836 | 0.0340 | 0.0191 | 60.64 | 91.70 | 98.21 |
| Australia | AERONET | 6426 | 516 | 63 | 0.488 | 0.654 | 0.363 | 0.1094 | 0.0827 | 0.0725 | 0.0711 | 0.0620 | 0.0563 | 59.96 | 59.88 | 71.43 |
| | TERRA | 284693 | 14588 | 1286 | 0.398 | 0.784 | 0.831 | 0.1091 | 0.0363 | 0.0188 | 0.0666 | 0.0261 | 0.0143 | 67.01 | 94.65 | 99.38 |
| Western Africa | AERONET | 2205 | 205 | 34 | 0.553 | 0.594 | 0.762 | 0.3180 | 0.2873 | 0.3357 | 0.2082 | 0.2029 | 0.2587 | 37.96 | 40.00 | 23.53 |
| | TERRA | 156392 | 10468 | 1028 | 0.501 | 0.769 | 0.849 | 0.1769 | 0.0706 | 0.0412 | 0.1198 | 0.0482 | 0.0242 | 51.83 | 88.01 | 97.57 |
| Southeast Asia | AERONET | 4134 | 504 | 74 | 0.405 | 0.542 | 0.488 | 0.2037 | 0.1447 | 0.1198 | 0.1274 | 0.0988 | 0.0821 | 50.17 | 56.15 | 60.81 |
| | TERRA | 402465 | 27058 | 2500 | 0.470 | 0.753 | 0.872 | 0.1730 | 0.0729 | 0.0342 | 0.109 | 0.0455 | 0.0198 | 57.25 | 87.01 | 97.96 |
| Eastern China | AERONET | 7396 | 927 | 118 | 0.513 | 0.551 | 0.356 | 0.3571 | 0.2355 | 0.1933 | 0.2038 | 0.1392 | 0.1382 | 40.10 | 49.84 | 50.00 |
| | TERRA | 241185 | 17324 | 1518 | 0.523 | 0.811 | 0.895 | 0.1646 | 0.0638 | 0.0302 | 0.1073 | 0.0435 | 0.0225 | 55.77 | 88.07 | 98.88 |
| Northeast Asia | AERONET | 9979 | 1178 | 142 | 0.569 | 0.593 | 0.367 | 0.4941 | 0.3249 | 0.2604 | 0.2924 | 0.2425 | 0.2202 | 35.17 | 29.54 | 21.13 |
| | TERRA | 78823 | 5485 | 467 | 0.553 | 0.872 | 0.965 | 0.1973 | 0.0636 | 0.0263 | 0.1201 | 0.0440 | 0.0198 | 56.48 | 87.77 | 98.29 |

| India | AERONET | 2208 | 203 | 32 | 0.521 | 0.462 | 0.534 | 0.2957 | 0.3015 | 0.3588 | 0.2049 | 0.2283 | 0.2862 | 32.11 | 24.63 | 15.63 |
|---|---|---|---|---|---|---|---|---|---|---|---|---|---|---|---|---|
| | TERRA | 179928 | 9564 | 862 | 0.526 | 0.815 | 0.915 | 0.1564 | 0.0599 | 0.0352 | 0.1089 | 0.042 | 0.0238 | 55.16 | 90.43 | 98.14 |

less than 0.01, the RMSE of 85% is less than 0.15, and the percentage falling within the EE of 67% is greater than 60%. More than 85% of stations fall within the EE is greater than 60% in Europe, North America, and Oceania, while 40-60% in South America, Africa, and Asia. The percentage of expected error is low in South and East Asia, and Central Africa, with some underestimation. Above 60% in Africa, Asia, North America, and Europe have a correlation coefficient greater than 0.4. The regions with lower correlation are the coastal regions of South America, eastern Africa, western Australia, northeastern North America, and northern Europe. Above 90% of the RMSE in Europe, North America, and Oceania have a correlation coefficient smaller than 0.15. High RMSE regions are in western North America, Asia, central South America, and central Africa.

Compared to AERONET daily AOD, the R of 74% stations is greater than 0.4, and the spatial distribution is similar to Terra's. The mean bias of 44% is less than 0.01, the RMSE of 68% is less than 0.15, and the percentage falling within the EE of 53% is greater than 60%. More than 70% of sites have a correlation coefficient greater than 0.4 in Africa, Asia, Europe, and North America. More than 57% of sites have an expected error percentage of over 60% in Europe, North America, and Oceania, except for Asia. Over 72% of sites have a RMSE less than 0.15. Except for Oceania and South America, over 71% of sites in other regions have MAE less than 0.01. Almost all sites in Asia show a negative bias, significantly underestimating. However, there is a significant overestimation in western North America and western Australia. Most sites in Asia falling within the expected error are less than 50%. High RMSE are in high emission and dust areas, such as Asia, India, and Africa.

The validation and comparison on the site scale show a limitation similar to the MODIS DT algorithm. In areas with high vegetation coverage, the AOD from visibility are better than those in bright areas. Although the correlation coefficients are high in high aerosol loading areas (Central South America, West Africa, India, Eastern China, Northeast Asia), there are significant differences in these areas with high RMSE values. As shown in Figure 6, some stations located in dusty and urban areas are overestimated or underestimated. Studies have shown that there is significant uncertainty in the MODIS retrievals in these regions, and the challenges of inversion algorithms are significant in bright surfaces (desert and snow covered areas) and urban surface of densely populated complex structures (Chu et al., 2002; Remer et al., 2005; Levy et al., 2010; Wei et al., 2019; Wei et al., 2020). In India, the elevation difference between AERONET site and meteorological station reached 0.7km may be a factor affecting the validation effect, as aerosol varies greatly with altitude. In eastern China, the complex urban surface, emission sources, and observations in different locations (AERONET site and meteorological station) may be the reasons for underestimation. At the same time, visibility stations in desert areas are sparse, and the spatial variability of dust aerosols is large, which also increases the difficulty to estimate VIS_AOD.

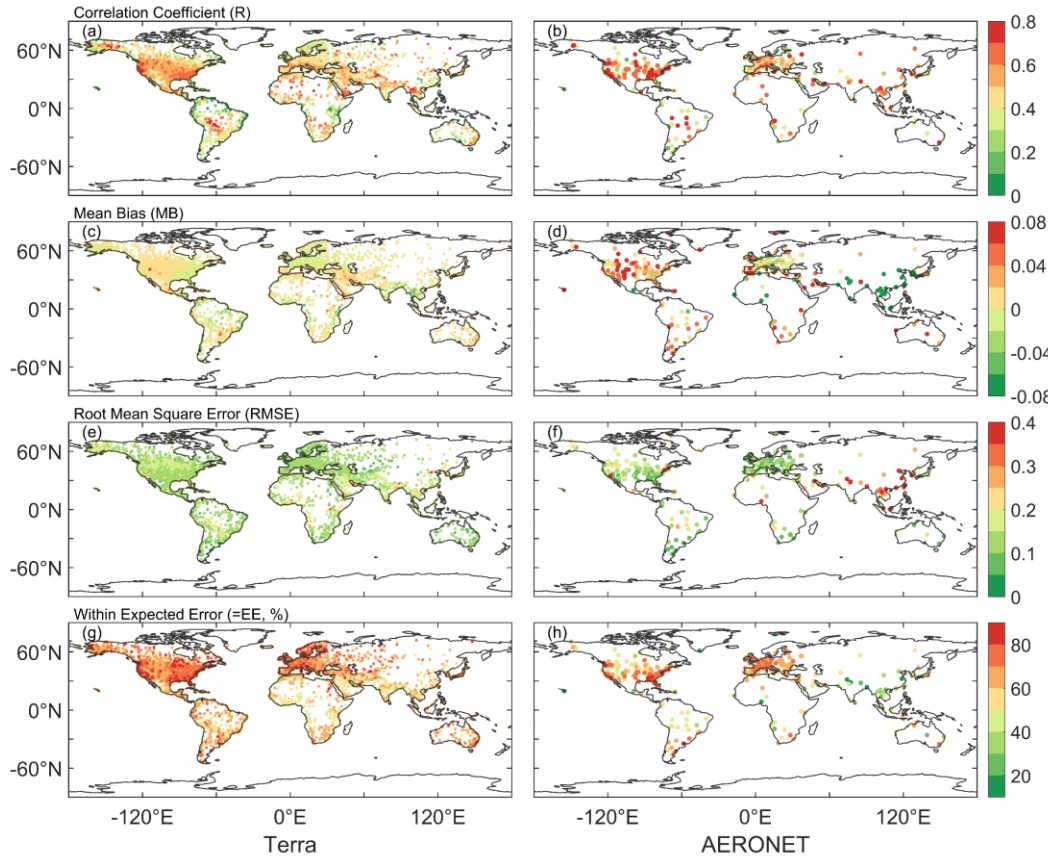

**Figure 8:** Validation of VIS_AOD against Terra and AERONET AODs at each site: (a–b) correlation (R), (c-d) mean bias (MB), (e-f) root mean square error (RMSE), (g-h) percentage (%) of VIS_AOD within the expected error envelopes.

### 3.3.4 Discussion and uncertainty analysis

The atmospheric visibility is a horizontal physical quantity, while AOD is a column-integrated physical quantity. We have linked the two variables together using machine a learning method, which partially compensates for the scarcity of AOD data. However, we have to face some limitations. Although the boundary layer height is considered, it is not sufficient. Pollutants such as smoke from biomass burning, dust, volcanic ash, and gas-aerosol conversion of sulfur dioxide to sulfate aerosols in the upper and lower troposphere can undergo long-range aerosol transport under the influence of circulation. The pollution transport and aerosol conversion processes above the boundary layer are still significant and cannot be ignored (Eck et al., 2023). Compared to surface visibility, bias occurs when the aerosol layer rises and affects AERONET measurements and MODIS retrievals. Therefore, it should be considered when using this data. If there were sufficient historical vertical aerosol measurements with high temporal and spatial resolution, the results of this data would be greatly improved. Although some studies use aerosol profiles from pollution transport models or assumed profiles as substitutes for observed profiles (Li et al., 2020; Zhang et al., 2020), the biases introduced by these non-observed profiles are still significant.

In machine learning, we use MODIS Aqua AOD as the target value for the model because the validation results for MODIS C6.1 product have a correlation coefficient of 0.9 or higher with AERONET AOD on the daily scale (Wei et al., 2019; Wei et al., 2020). Compared to AERONET, MODIS AOD provides more sample data with a high global coverage. However, apart from

662 modeling errors, the systematic biases and uncertainties of MODIS Aqua AOD cannot be ignored
(Levy et al., 2013; Levy et al., 2018; Wei et al., 2019). Averaging over time scale can reduce
representation errors effectively, and emission sources and orography can increase representation
errors (Schutgens et al., 2017). Therefore, the strong correlation at monthly and annual scales
indicates a substantial reduction in errors. This is also one of the reasons why this dataset shows
stronger correlation with Terra AOD and weaker correlation with AERONET in validation.
The spatial matching between meteorological stations and AERONET sites may cause some biases.
AERONET sites are usually not co-located with meteorological stations in terms of elevation and
horizontal distance, this is another reason for the weak correlation between VIS_AOD and
AERONET AOD. The meteorological stations are located at the airport. Different horizontal
distances may result in meteorological stations and AERONET sites being located on different
surfaces (such as urban, forest, mountainous). Differences in site elevation significantly impact the
relationship between AOD and measured visibility. When the AERONET site is at a higher elevation
than the meteorological station, there may be fewer measurements of aerosols over the sea at the
AERONET site.
Different pollution levels and station elevation affect the AOD derived from visibility. The elevation
difference and distance between meteorological stations and AERONET sites also have an impact
on the validation results. Therefore, the error and performance of different AERONET AOD values,
station elevation, and distance are analyzed.
As the AOD increases, the variability of bias also increases in Figure 9 (a). Almost all mean bias
values are within the envelope of EE, except for 1.1-1.2 and 1.5-1.6. The average bias is 0.015
(AOD <0.1), with 83% of data within the EE envelopes. The mean bias is -0.0011 (AOD,0.1-0.2),
with 54% within the EE envelopes. The mean bias is negative (AOD, 0.3-1.0), with 20%-40%
falling within the EE envelopes. There is a positive bias (AOD, 1.1, 1.4 and >1.6), and there is a
negative bias at 1.2-1.3 and 1.5-1.6. The results indicate that as pollution level increases, the
negative mean bias becomes significant and the underestimation increases.
The contribution of particulate matter near the ground to the column aerosol loading is significant.
The elevation of the site affects the measurement of column aerosol loading in Figure 9 (b). There
is a negative bias in the low elevation (<=0.5km) with a percentage of 60%-64% falling within the
EE envelopes and a positive bias in high elevation (0.5-1.2km) with a percentage of 50%-65%
falling within the EE envelopes. The percentage significantly decreases (>1.2km), and the average
bias increases. Therefore, the elevation of AERONET's site will cause bias in validation, and. the
uncertainty greatly increases in high elevation.
Due to the elevation difference between the meteorological station and AERONET site in the
vertical direction, the uncertainty caused by elevation differences of site was analyzed in Figure 9
(c). When the elevation difference is negative (the elevation of the meteorological station is lower
than that of the AERONET station), there is a significant positive bias. When the difference is
positive, the mean bias approaches 0 or is positive. The percentage is greater than 60% (-0.5 km-
0.5km). The positive mean bias is greater than the negative mean bias, and the uncertainty greatly
increases when the elevation of meteorological stations is lower than that of AERONET sites. It
indicates that the contribution of the near surface aerosol to the column aerosol loading is significant
and cannot be ignored.
The spatial variability of aerosols is significant. Meteorological stations and AERONET sites are
not collocated, resulting in a certain distance in spatial matching. In this study, the upper limit of

distance is 0.5 degree. Figure 9 (d) shows the error of the distance between stations, where the degree is converted to the distance at WGS84 coordinates. The bias does not change significantly with increasing distance. The average bias is around 0, with the maximum positive mean bias (0.0322) at a distance of 2km and the maximum negative mean deviation (-0.0323) at 6km. The median is almost positive, except at 5km and 6km. The percentage falling within the EE envelopes is over 50%, with the maximum percentage (66%) at 3km and the minimum (62%) at 2km.

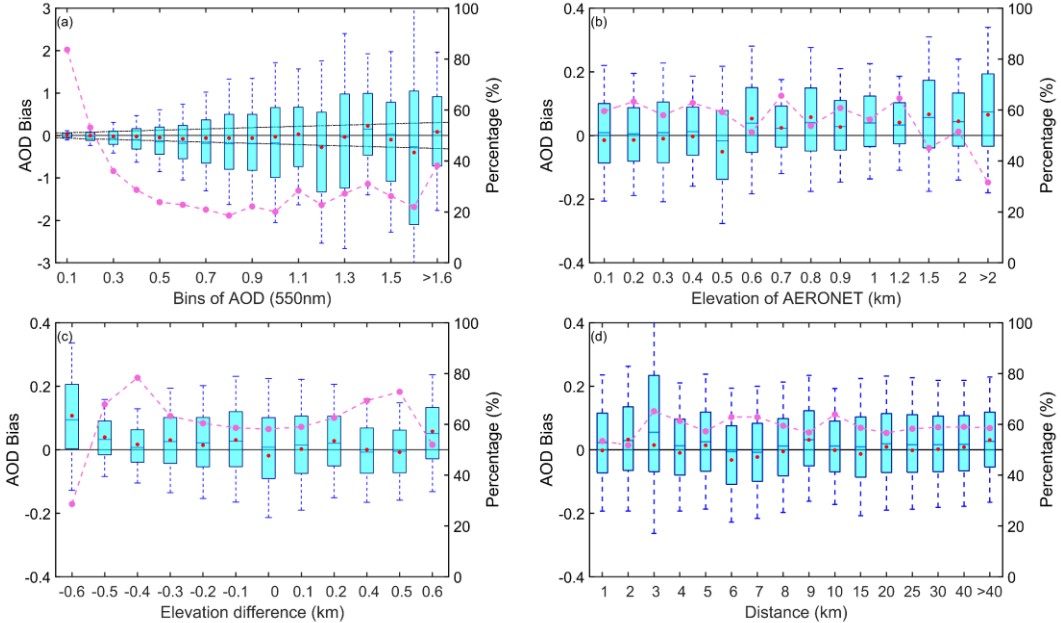

**Figure 9:** Box plots of AOD bias and the percentage falling within the EE envelopes (curves): (a) AERONET AOD levels, (b) elevation of AERONET sites, (c) elevation difference between meteorological stations and AERONET sites, (d) distance (km) between meteorological stations and AERONET sites. The black horizontal line represents the zero bias. For each box, the upper, lower, and middle horizontal lines, and whiskers represent the AOD bias 75th and 25th percentiles, median, and 1.5 times the interquartile difference, respectively. The black solid lines represent the EE envelopes ($\pm(0.05+0.15*AOD_{AERONET})$). No site with a difference of +0.3km (x-axis label without 0.3) in (c).

**3.4 Gridded visibility-derived AOD**

**3.4.1 Uncertainty of gridded AOD**

We calculate the width of the 95% CI for gridded AOD. Figure 10 (a-b) shows the spatial distribution and frequency of the 95% CI from 1980 to 2021. In areas with dense visibility stations, the kriging variance is low, the width of 95% CI is small, and the uncertainty of the gridded AOD is low. In areas with sparse visibility stations, the width is large, and the uncertainty is high. The uncertainty of approximately 43% of the grids is less than 0.03, and nearly 80% has an uncertainty less than 0.06. Approximately 7% of the grids have an uncertainty larger than 0.1. Regions with low uncertainty are mainly located in North America (<60°N), Europe, Western and Southern Asia, Eastern China, and South America. Regions with high uncertainty are found in high-latitude areas (e.g., Siberia), high-altitude regions (e.g., Tibetan Plateau), and desert areas (such as the Sahara

Desert, Taklamakan Desert, and Australian deserts).

Uncertainty also exhibits seasonal variations, as shown in Figures (c-f). The percentage of grid cells
with uncertainty less than 0.06 is 63%, 84%, 77%, and 86% in DJF, MAM, JJA, and SON,
respectively. Compared to other seasons, uncertainty increases significantly in high-latitude regions,
Africa, northern Asia, Oceania, and eastern South America during DJF. In JJA, the distribution of
uncertainty is similar to DJF, but the uncertainty decreases. In MAM and JJA, there is higher
confidence, with a small number of grid cells having large uncertainty (>0.1), primarily concentrated
in high-latitude regions.

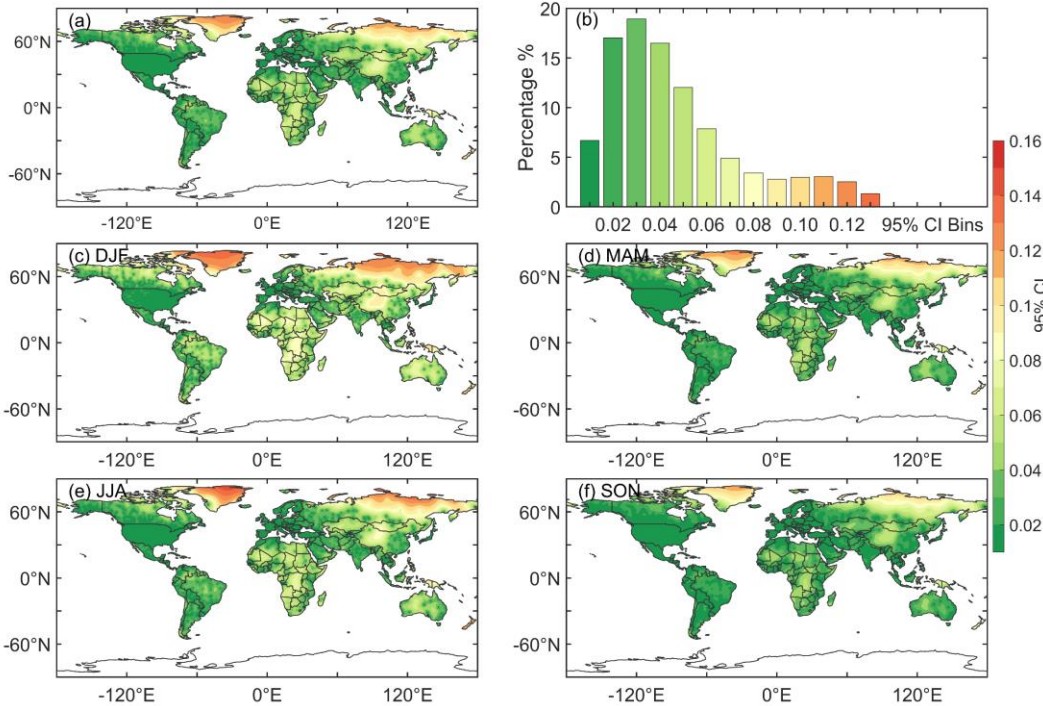

**Figure 10:** The spatial distribution (a) and frequency (b) of the 95% confidence interval (CI) from
1980 to 2021 The spatial distribution of the width of the 95% CI for each season (c-f). Bins of 95%
CI are from 0 to 0.15 with an interval of 0.01. DJF represents December and next January and
February. MAM represents March, April, and May. JJA represents June, July, and August. SON
represents September, October, and November.

**3.4.2 Comparison with Aqua/Terra MODIS AOD**

Figure 11 shows the gridded AOD based on ordinary kriging interpolation with the area-weighted
method and compares the multi-year spatial, zonal, and meridional distributions of AOD with Aqua
and Terra AOD over land from 2003 to 2021. The VIS_AOD is 0.157±0.073 over land, which is
almost equal to the Aqua (0.152±0.084) and Terra (0.154±0.088) AOD values with relative biases
of 3.3%, and 1.9%, respectively. In order to compare the spatial correlation, Aqua and Terra MODIS
AOD are averaged to the 0.5-degree resolution. In the heatmap (Figure 12), the R of VIS_AOD and
Aqua AOD is 0.798, the RMSE is 0.049 with a bias of 32% compared to the mean, and the MAE is
0.008, with a bias of 5% compared to the mean. Compared to Terra AOD, the R is 0.787, and the
RMSE is 0.051, with a bias of 33% compared to the mean, and the MAE is 0.005, with a bias of 3%
compared to the mean. The R between Aqua and Terra AOD is 0.980. The R values between

VIS_AOD and Aqua and Terra AOD are 0.995 and 0.990 for the zonal distribution and 0.986 and 0.897 for the meridional distribution, respectively. In the low aerosol loading region, VIS_AOD exhibits a little overestimation. Whether in meridional or zonal distribution, the peak and valley regions are basically consistent (Tian et al., 2023). Due to the limitations of satellite inversion algorithms, a bias appears on the bright surface, especially in northern North America with extensive snow cover (Levy et al., 2013). All above results suggest that the gridded AOD is consistent with satellite retrievals in spatial distribution.

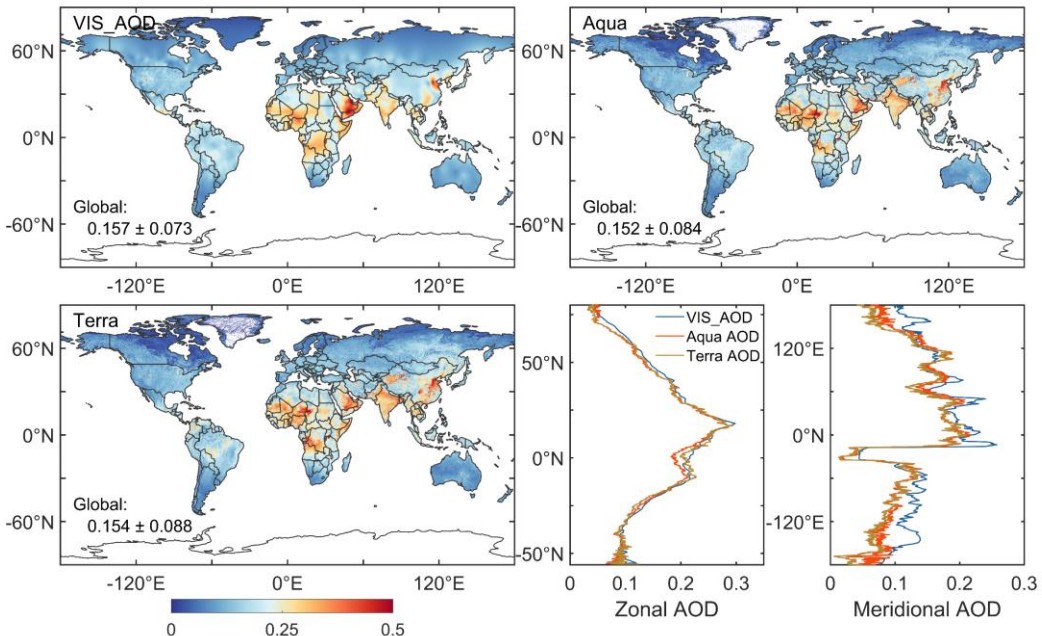

**Figure 11:** The spatial, zonal and meridional distributions of the multi-year mean VIS_AOD, Aqua AOD, and Terra AOD over land from 2003 to 2021.

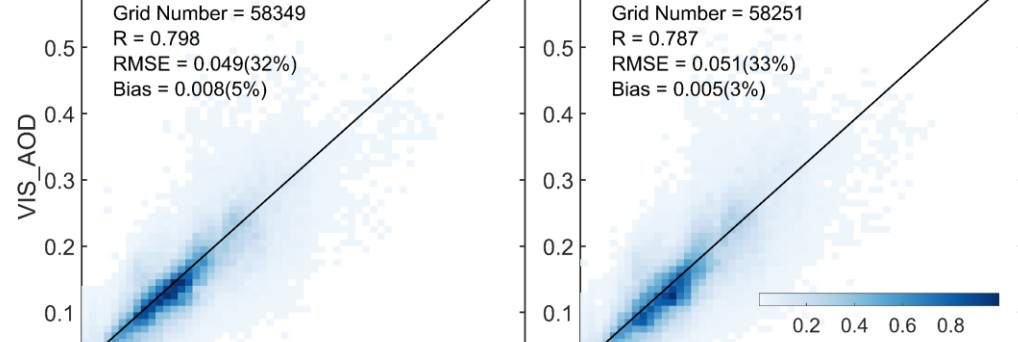

**Figure 12:** Heatmap of multi-year mean gridded VIS_AOD and Aqua AOD and Terra AOD during 2003-2021. Terra and Aqua AOD are averaged onto a grid of 0. 5°.

**3.5 Interannual variability and trend of visibility-derived AOD over global land**

The spatial distribution of multi-year average AOD from 1980 to 2021 over land is shown in Figure 13 (a). The mean AOD of land (-60-85°N), Northern Hemisphere (NH, 0-85°N), and the Southern Hemispheres (SH, -60-0°N) is 0.161 ± 0.074, 0.158 ± 0.076, and 0.173 ± 0.059, respectively. The AOD values of Africa, Asia, Europe, North America, Oceania, and South America are 0.241, 0.222, 0.110, 0.111, 0.129 and 0.117, respectively.

Due to the influence of geography, atmospheric circulation, population, and emissions, the AOD varies in different latitudes. Figure 14 illustrates the multi-year average AOD in different latitude ranges for land, the NH, and the SH from 1980 to 2021. Within [-20, 20°N], the global average AOD reaches its maximum (0.234), and the maximum AOD NH is 0.256 in [0, 20°N]. The highest AOD in SH is 0.217 in in [-15, 0°N]. The average AOD in SH rapidly decreases from -15°N to -35°N. In NH, AOD is generally greater than in SH from 5°N to 65°N. When, the latitude is greater than 70°N, the NH's AOD is smaller than the SH's.

There are many regions of high AOD values occur in NH, with the distribution of population density. Approximately 7/8 of the global population resides in the NH, with 50% concentrated at 20°N-40°N (Kummu et al., 2016), indicating a significant impact of human activities on aerosols. The highest AOD values are observed near 17°N, including the Sahara Desert, Arabian Peninsula, and southeastern India, suggesting that in addition to anthropogenic sources, deserts also play a crucial role in aerosol emissions. Lower AOD regions of the SH are from 25°S to 60°S, encompassing Australia, southern Africa, and southern South America, indicating lower aerosol burdens in these areas. Additionally, North America also exhibits low aerosol loading. Chin et al. (2014) analyzed the AOD over land from 1980 to 2009 with the Goddard Chemistry Aerosol Radiation and Transport model, which is similar to the visibility-derived AOD. The spatial distribution is consistent with the satellite results (Remer et al., 2008; Hsu et al., 2012; Hsu et al., 2017; Tian et al., 2023). The AOD and extinction coefficient retrieved from visibility show a similar distribution at global scale, with a correlation coefficient of nearly 0.6 (Mahowald et al., 2007). Similar global (Husar et al., 2000; Wang et al., 2009) and regional (Koelemeijer et al., 2006; Wu et al., 2014; Boers et al., 2015; Zhang et al., 2017; Zhang et al., 2020) spatial distributions have been reported.

AOD loadings exhibit significant seasonal variations worldwide, particularly over land. In this study, a year is divided into four parts: December-January-February (DJF), March-April-May (MAM), June-July-August (JJA), and September-October-November (SON), corresponding to winter (summer), spring (autumn), summer (winter), and autumn (spring) in NH (SH), respectively. Figure 13 (b-e) also depicts the spatial distribution of seasonal average AOD over land from 1980 to 2021. The global AOD in DJF, MAM, JJA, and SON is 0.158±0.062, 0.162±0.081, 0.175±0.093, and 0.153± 0.070, respectively. The standard bias of AOD in JJA and MAM are greater than those in DJF and SON. AOD exhibits seasonal changes, with the highest in JJA, followed by MAM, DJF, and SON. From 1980 to 2021, the seasonal AOD in NH is 0.152±0.064 (DJF), 0.161±0.088 (MAM), 0.176±0.090 (JJA), and 0.144±0.060 (SON), and in SH is 0.184±0.041 (DJF), 0.166±0.044 (MAM), 0.169±0.072 (JJA), and 0.19±0.060 (SON).

In NH, the AOD ranking from high to low in season is summer > spring > winter > autumn. In SH, the AOD ranking from high to low in season is spring > summer > winter > autumn. The highest AOD is observed during JJA in NH, while in SH, the peak occurs during SON. The occurrence of

high AOD values is highly associated with the growth of hygroscopic particle and the photochemical reaction of aerosol precursors under higher relative humidity in Asia (JJA) (Remer et al., 2008) and Europe such as Russia (JJA), and biomass burning in South America (SON), Southern Africa (SON), and Indonesia (SON) (Ivanova et al., 2010; Krylov et al., 2014). On the other hand, the lowest global AOD values are observed during autumn, which may be attributed to the weakening of monsoon systems (Li et al., 2016; Zhao et al., 2019).

In addition to the spatial characteristics of AOD, the temporal variations in AOD have also been of great interest due to the significant relationship between aerosols and climate change. Figure 13 (f) shows the temporal trends of annual average AOD (** represents passing the significance test, $p<0.01$) over the global land, the SH and the NH during 1980-2021. The global land, NH, and SH trends demonstrate decreasing trends of AOD with values of -0.0026/10a, -0.0018/10a, and -0.0059/10a, respectively, with all passing the significance test with a confidence level of 95%. Notably, the declining trend is much greater in the SH than in the NH. It may be related to the decrease in the frequency of sandstorms and wildfires and the increase in precipitation, such as in Australia. Two AOD peaks in 1983 and 1994 and two AOD valleys in 1980 and 1990 are observed before 2000. The two AOD peaks may be attributed to large volcanic eruptions, which has been confirmed by previous studies. The volcanic eruptions and their associated fires of the El Chichón volcano in Mexico in 1982 (Hirono and Shibata, 1983) and Mount Pinatubo in the Philippines in 1991(Tupper et al., 2005) resulted in elevating global AOD levels in the following years. The AOD recovery to the previous low levels after volcanic eruptions takes approximately 10 years (Chazette et al., 1995; Sun et al., 2019). This further indicates the efficiency of our data capturing the volcanic eruption emission features.

Due to the influence of geography, atmospheric circulation, population, and emissions, the trend of global aerosols varies in different latitude Figure 14 illustrates the multi-year average AOD in different latitude ranges for land, the NH, and the SH from 1980 to 2021. Within [-20, 20°N], the global average AOD reaches its maximum (0.234), and the maximum AOD NH is 0.256 in [0, 20°N]. The highest AOD in SH is 0.217 in in [-15, 0°N]. The average AOD in SH rapidly decreases from -15°N to -35°N. In NH, AOD is generally greater than in SH from 5°N to 65°N. When, the latitude is greater than 70°N, the NH's AOD is smaller than the SH's, which may be related to low emission intensity and low population density in high latitude areas.

The seasonal trends of AOD during 1980-2021 at the global and hemispheric scales are shown in Figure 13 (g-j). The global AOD shows a decreasing trend in all seasons (-0.002~-0.003/10a). The large declining trends are observed in JJA and SON, with decreasing trend values of -0.003/10a and -0.0029/10a, respectively. DJF and MAM follow with decreasing trend values of -0.0026/10a and -0.0022/10a, respectively, all passing the significance test ($p<0.01$). For the NH, the AOD trends in different seasons are -0.0030/10a (DJF), -0.0006/10a (MAM), -0.0005/10a (JJA), and -0.0034/10a (SON). DJF and SON pass the significance test ($p<0.01$), while MAM and JJA do not. In the SH, the trends are as follows: -0.0011/10a (DJF), -0.0085/10a (MAM), -0.0131/10a (JJA), and -0.0009/10a (SON). Interestingly, in contrast to the NH, MAM and JJA pass the significance test, while DJF and SON do not. The largest declining season in the NH is winter, while in the SH, it is summer. The decreasing trend in the SH is more than four times greater than that in the NH, particularly before the year 2000. While both the global and SH AOD exhibit a decreasing trend since 2005, the NH has shown a significant increase in winter AOD, leading to an overall increasing

trend. Moreover, the NH shows an increasing trend of 0.004/10a from 2005 to 2021.

Annual $SO_2$ emissions increased from 9.4 to 15.3 TgS from 2000 to 2005, which ultimately ended
up as sulfate aerosols, leading to a significant increase in sulfate aerosols (Hofmann et al., 2009).
More relevantly, the frequent volcanic eruptions in tropical regions from 2002 to 2006, combined
with seasonal circulation patterns during winter, led to the transport of aerosol particles to higher
latitudes (Hofmann et al., 2009; Vernier et al., 2011; Sawamura et al., 2012; Andersson et al., 2015).

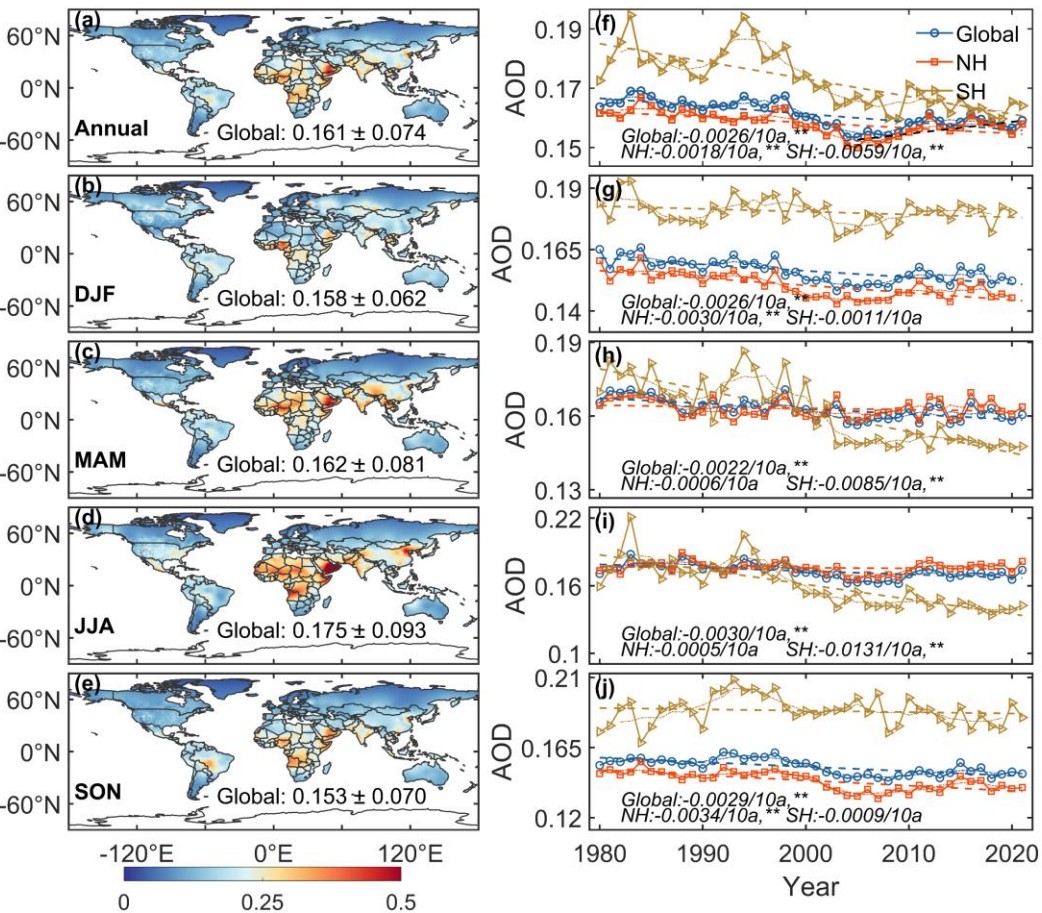

**Figure 13:** The multi-year averages of VIS_AOD from 1980 to 2021. Global land (circle), northern
hemisphere (NH,0-85°N) (triangle) and southern hemisphere (SH,0-60°S) (square) annual and
seasonal AOD. The symbol, **, represents that the test passed at a significance level of 0.01. DJF
represents December and next January and February. MAM represents March, April, and May. JJA
represents June, July, and August. SON represents September, October, and November.

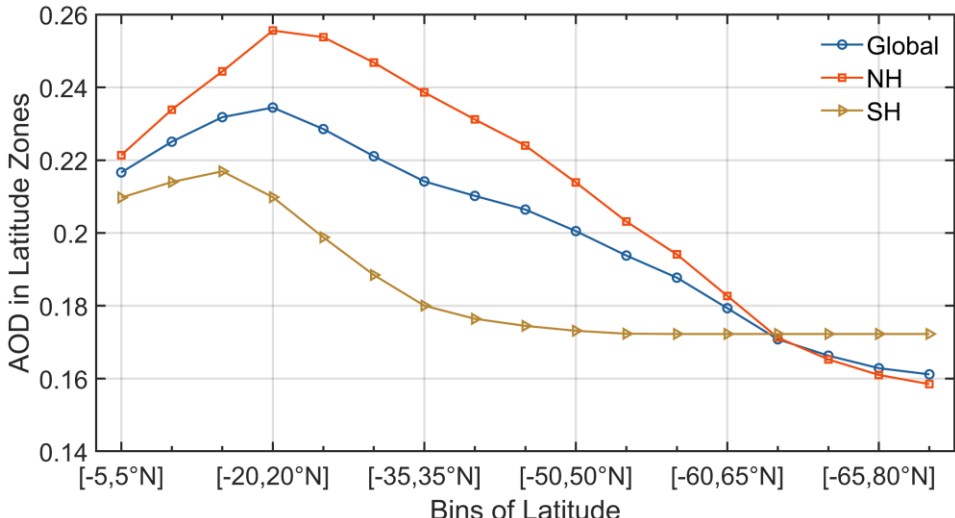

**Figure 14:** The global land (blue), northern hemisphere's (red) and southern hemisphere's (yellow) multi-year average VIS_AOD from 1980 to 2021 in different latitude zones. The latitude range is from -60 to 85°N, with a bin of 5°.

**3.6 Interannual variability and trend of visibility-derived AOD over regions**

The distribution of AOD over global land exhibits significant spatial heterogeneity. Large variations in aerosol concentrations exist among different regions, leading to a non-uniform spatial distribution of AOD globally. Accurately assessing the long-term trends of aerosol loading is a key for quantifying aerosol climate change, and it is crucial for evaluating the effectiveness of measures implemented to improve regional air quality and reduce anthropogenic aerosol emissions.

To analyze the spatiotemporal characteristics and trends of AOD in different regions, we selected 12 representative regions that are influenced by various aerosol sources(Wang et al., 2009; Hsu et al., 2012; Chin et al., 2014), such as desert, industry, anthropogenic emissions, and biomass burning emissions, which nearly cover the most land and are densely populated regions (Kummu et al., 2016). These representative regions are Eastern Europe, Western Europe, Western North America, Eastern North America, Central South America, Western Africa, Southern Africa, Australia, Southeast Asia, Northeast Asia, Eastern China, and India, as shown in Figure 1. We use multi-year average and seasonal average AOD to evaluate aerosol loadings (Figure 15), the annual average of monthly anomalies to analyze interannual trends (Figure 16), and the seasonal average to analyze seasonal trends (Figure 17) in 12 regions from 1980 to 2021.

We can see some differences between VIS_AOD and MODIS AOD. In addition to model errors, the spatial matching between meteorological stations and MODIS, terrain, surface coverage, and station altitude will also bring errors. When particle transport and photochemical reactions occur above the boundary layer, visibility cannot capture the feature, which will also increase the uncertainty. However, bias is inevitable and can only be kept as small as possible. From the trend, they have similar changing characteristics, especially on monthly and yearly scales.

Figure 15 shows the regions with high AOD level from 1980 to 2021 (multi-year average AOD > 0.2) are in West Africa, Northeast Asia, Eastern China, and India. The AOD values in Eastern North

America, Central South America, South Africa, and Southeast Asia range from 0.15 to 0.2. The AOD values in Eastern Europe, Western Europe, Western North America, and Australia are less than 0.15.

Europe is an industrial region with a low aerosol loading region, and the multi-year average AOD in Eastern Europe (0.144±0.007) is higher than that in Western Europe (0.139±0.003) during 1980-2021. Eastern Europe shows a greater downward trend in AOD (-0.0041/10a) compared to Western Europe (-0.0021/10a). The highest AOD is observed in JJA, the dry period when solar irradiation and boundary layer height increase, with Eastern Europe at 0.161 and Western Europe at 0.162, which could be due to increases in secondary aerosols, biomass burning, and dust transport from the Sahara (Mehta et al., 2016). However, there are seasonal variations. In Eastern Europe, the seasonal AOD ranking from high to low is JJA (0.161) > DJF (0.147) > MAM (0.138) > SON (0.131), while in Western Europe, it is JJA (0.162) > MAM (0.140) > SON (0.136) > DJF (0.117). The differences among seasons are larger in Western Europe. AOD in Eastern Europe shows declining trends in all seasons, while it does not pass the significance test in MAM. Among four seasons, SON has the largest decline trend of AOD (-0.0058/10a). In Western Europe, DJF, JJA, and SON exhibit declining trends of AOD that pass the significance test, while the MAM shows a significant increase trend of AOD (0.0022/10a), which may be due to eruptions of the Eyjafjallajökull volcano in Iceland in spring 2010 (Karbowska and Zembrzuski, 2016). Both Western and Eastern Europe experienced increasing trends in AOD during the period of 1995-2005, with Western Europe showing a greater increase. However, after 2000, the decline rate accelerated in both regions. The downward trend in Europe is attributed to the reduction of biomass burning, anthropogenic aerosols, and aerosol precursors (such as sulfur dioxide)(Wang et al., 2009; Chin et al., 2014; Mortier et al., 2020).

North America is also an industrial region with a low aerosol loading. The average AOD values for Eastern and Western North America during 1980-2021 are 0.153±0.004 and 0.131±0.005, respectively, with the Eastern region being higher than the Western region by 0.022. From 1980 to 2021, both Eastern (-0.0021/10a) and Western North America (-0.0009/10a) show a downward trend; however, the decline in the Western region is not statistically significant. And the trend is -0.0172/10a from 1995 to 2005 and 0.0096/10a from 2005 to 2021.The average AOD values in DJF, MAM, JJA, and SON in Western North America are 0.1367, 0.1286, 0.1457, and 0.114, respectively, compared to 0.137, 0.145, 0.1913, and 0.138 in Eastern North America. The lowest AOD values of 12 regions during DJF and SON are observed in Western North America (Remer et al., 2008). Specifically, in the Western region, there is a consistent increasing trend during MAM (0.004/10a) from 1980 to 2021, while JJA and SON also show an increase after 2000, except for DJF (-0.0032/10a). In contrast, the AOD trends in the Eastern region remain unchanged during the period 1980-2021, except for MAM, which shows a stable increasing trend (0.0033/10a), while DJF, JJA, and SON exhibit decreasing trends (-0.0023/10a, -0.0040/10a, -0.0053/10a, respectively). In the Western region, the annual mean AOD started to increase after 2005, while in the Eastern region, the increase was not significant. The upward trend may be due to low rainfall and increased wildfire activities (Yoon et al., 2014). The decrease in AOD in Eastern North America is related to the reduction of sulfate and organic aerosols, as well as the decrease in anthropogenic emissions caused by environmental regulations (Mehta et al., 2016).

Central South America is a relatively high aerosol loading region, sourced from biomass burning, especially in SON (Remer et al., 2008; Mehta et al., 2016), with a multi-year average AOD of 0.192±0.017. There is a clear downward trend (-0.0100/10a) from 1980 to 2021, which is slightly greater than the trend (-0.0090/10a) from 1998 to 2010 (Hsu et al., 2012) and AOD decreased from 1980 to 2006 (Streets et al., 2009) and from 2001 to 2014 (Mehta et al., 2016). Although DJF (0.199) and SON (0.226) have higher values compared to MAM (0.180) and JJA (0.163), the large declining trends are observed in MAM (-0.0126/10a) and JJA (-0.0167/10a). It indicates that although AOD has decreased overall, the aerosol loading caused by seasonal deforestation and biomass combustion is still large(Mehta et al., 2016).

Africa is also one of the regions with a high aerosol loading worldwide. In West Africa, the average AOD is 0.275±0.012 during 1980-2021, and the annual AOD shows a downward trend (-0.0008/10a, $p>0.05$). The world's largest desert (Sahara Desert) is in West Africa, with much dust aerosol discharged. AOD values in all seasons are above 0.25, with JJA (0.301) and MAM (0.300) reaching 0.3, and DJF and SON being 0.252 and 0.250, respectively. The AOD in DJF (-0.0135/10a, $p<0.01$) and SON (-0.0026/10, $p>0.05$) exhibit decreasing trends, while JJA (0.0088/10a, $p<0.01$) and MAM (0.0037/10a, $p>0.05$) show an opposite trend. The multi-year average AOD in South Africa is 0.177±0.020, lower than that of West Africa. The annual mean AOD in South Africa shows a significant decrease (-0.0096/10a). The AOD values range from 0.12 to 0.2 during 2000-2009, dominated by fine particle matter from industrial pollution from biomass and fossil fuel combustion (Hersey et al., 2015). The average AOD values in DJF, MAM, JJA, and SON are 0.189, 0.162, 0.147, and 0.210, respectively. JJA (-0.0268/10a, $p<0.01$), MAM (-0.0126/10a, $p<0.01$) and SON (-0.0001/10a, $p>0.05$) exhibit a downward AOD trend, while DJF (0.0006/10a, $p>0.05$) shows an upward trend. AERONET and simulation results also show a decreasing trend of AOD (Chin et al., 2014).

Australia is a region with a low aerosol loading. The multi-year mean AOD is 0.127±0.014 during 1980-2021. The AOD ranges from 0.05 to 0.15 from AERONET during 2000-2021, and dust and biomass burning are important contributors to the aerosol loading (Yang et al., 2021a). There is a downward trend of AOD (-0.0081/10a, $p<0.01$), which may be related to a decrease in dust and biomass burning (Yoon et al., 2016; Yang et al., 2021a). In addition, research has shown that the forest area in Australia has increased sharply since 2000 (Giglio et al., 2013), surpassing the forest fire area of the past 14 years. The seasonal average of AOD in MAM, JJA, SON, and DJF are 0.122, 0.108, 0.125, and 0.151. The AOD in JJA is the lowest among all seasons and regions. The highest AOD is in DJF with an increasing trend (0.0056/10a, $p<0.01$), while the trends during MAM, JJA and SON are -0.0096/10a ($p<0.01$), -0.0231/10a ($p<0.01$) and -0.0042/10a ($p<0.01$), respectively. Ground-based observations and satellite retrievals indicate that wildfires, biomass burning and sandstorms lead to high AOD in DJF and SON. The low AOD of MAM and JJA is due to a decrease in the frequency of sandstorms and wildfires and an increase in precipitation (Gras et al., 1999; Yang et al., 2021a; Yang et al., 2021b).

Asia is also a high aerosol loading area with various sources. In Southeast Asia, the multi-year average AOD is 0.177 during 1980-2021 with a downward trend of AOD (-0.0003/10a, $p>0.05$). It is also a biomass-burning area. The seasonal average AOD ranking from high to low is JJA (0.207) > MAM (0.183) > DJF (0.169) > SON (0.149). The trends in DJF (-0.0035/10a, $p<0.05$), JJA (-0.0007/10a, $p>0.05$) and SON (-0.0021/10a, $p>0.05$) are opposite to MAM (0.0050/10a, $p<0.01$).

Southeast Asia has no clear long-term trend in estimated AOD or observed surface solar radiation (Streets et al., 2009). In Northeast Asia, the multi-year average AOD is 0. 222 during 1980-2021, with no significant temporal trend. The seasonal AOD values are 0.252 in MAM, 0.215 in DJF, 0.212 in SON and 0.209 in JJA. AOD in MAM is significantly higher than other seasons, which may be related to sandstorms in East Asia, and the reason for the high AOD in winter may be related to the transportation. The trends of AOD in DJF (-0.0025/10a, $p>0.05$), MAM (0.0031/10a, $p>0.05$), JJA (0) and SON (-0.0006/10a, $p>0.05$) are not significant. In Eastern China, the multi-year average AOD is 0.233, with an increasing trend (0.0071/10a, $p<0.01$). The trend is 0.0151/10a from 1980 to 2006 and -0.0469/10a from 2006 to 2021.The seasonal average AOD ranking from high to low is JJA (0.284), MAM (0.234), SON (0.230) and DJF (0.183). The AOD trends in DJF (0.0093/10a, $p<0.01$), MAM (0.0092/10a, $p<0.01$), JJA (0.0038/10a, $p>0.05$) and SON (0.0065/10a, $p<0.05$) are all positive but the trend in JJA does not pass the significance test. We can see that there are three stages of changes in AOD: 1980-2005, 2006-2013 and 2014-2021. In the first stage, AOD increased steadily. In the second stage, AOD maintained a larger positive anomaly accompanied by oscillations. The third stage experienced a rapid decline, reaching the level of the 1980s by 2021. The increasing trend of AOD before 2006 may be due to the significant increase in industrial activity, and after 2013, the significant decrease is closely related to the implementation of air quality-related laws and regulations, along with adjustments in the energy structure (Hu et al., 2018; Cherian and Quaas, 2020).

India is a high aerosol loading area. The multi-year average AOD is 0.255, with an upward trend (0.0096/10a, $p<0.01$) from 1980 to 2021. Dust and biomass burning has an influence on AOD level. There are three stages: 1980-1997 (0.0032/10a, $p<0.01$), 1997-2005 (-0.0420/10a, $p<0.01$), 2005-2021 (0.0481/10a, $p<0.01$). Although the trend is downward in the second stage, the lager positive trend is in the third stage. The seasonal average AOD values are 0.237 in DJF, 0.258 in MAM, 0.269 in JJA, and 0.256 in SON. The largest AOD is in JJA. In winter and autumn, it affected by biomass burning, and in spring and summer, it is also affected by dust, transported from the Sahara under during the monsoon period (Remer et al., 2008). The trends in DJF (0.0152/10a, $p<0.01$), MAM (0.0091/10a, $p<0.01$), JJA (0.0025/10a, $p>0.05$), and SON (0.0107/10a, $p<0.05$) are positive. There largest trend is in winter.

To summarize, there are significant differences in the spatial distribution, annual trends, and seasonal trends of AOD across different regions from 1980 to 2021. The high aerosol loadings from 1980 to 2021 are in West Africa, India and Asia, and low aerosol loading regions are in Europe, Western North America, and Australia. Eastern China and India show an increasing trend, Southeast Asia and Northeast Asia show no significant trend, and the other regions show downward trends. However, not all regional seasonal trends are consistent with their annual trends. The results in this study have supplemented the long-term trend and distribution of AOD over land.

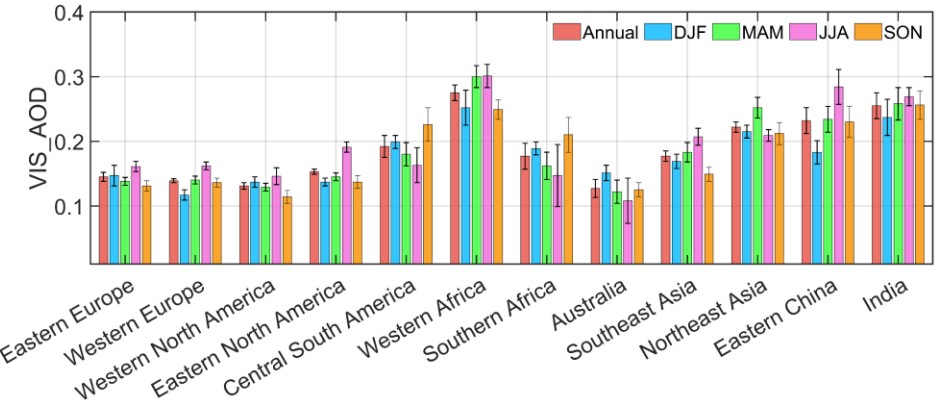

1019

**Figure 15:** Annual and seasonal averages of AOD in 12 regions during 1980-2021.

1021

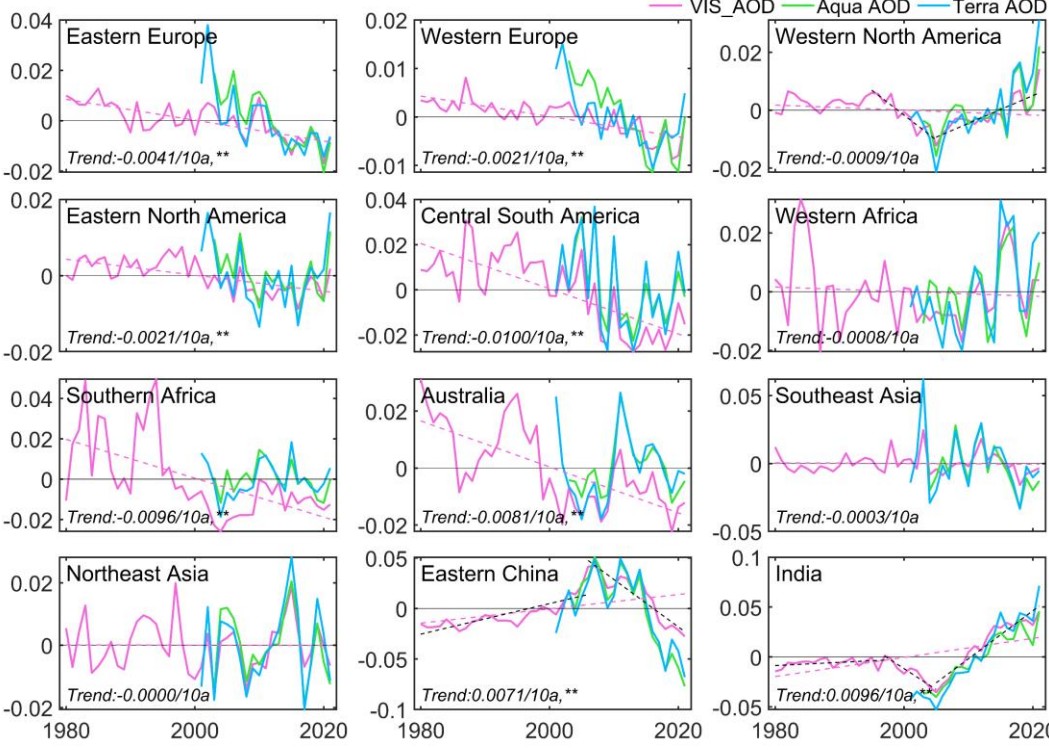

**Figure 16:** Annual averages of monthly anomaly gridded VIS_AOD (pink line), Aqua (green line), and Terra (blue line) MODIS AOD in 12 regions. The dotted line is the trend line.

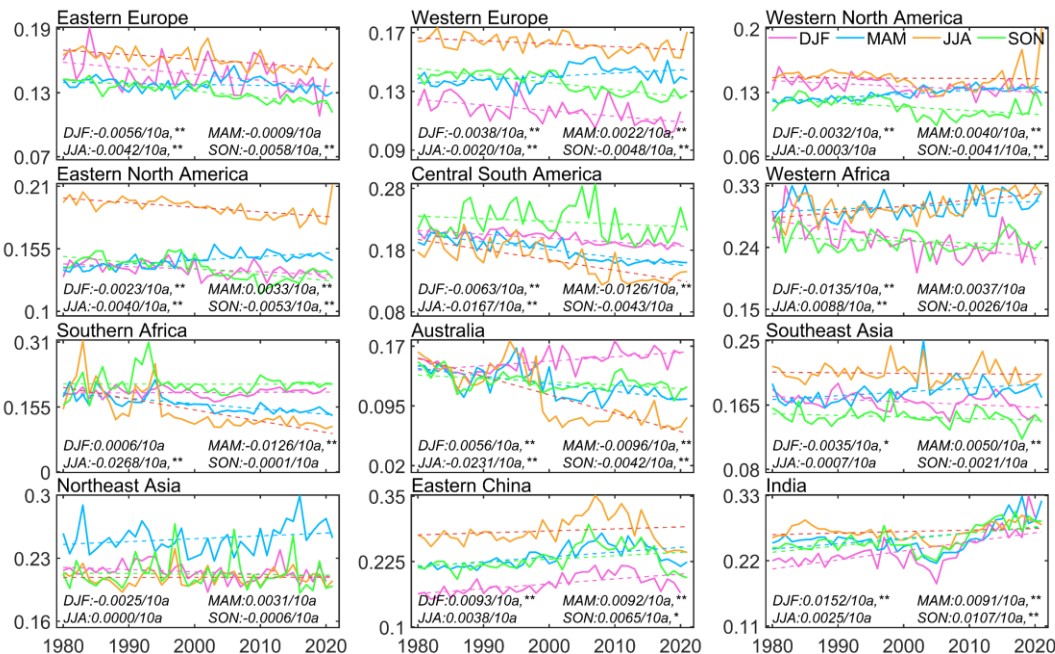

**Figure 17:** Seasonal averages of gridded VIS_AOD during 1980 to 2021 in 12 regions (Eastern Europe, Western Europe, Western North America, Eastern North America, Central South America, Western Africa, Southern Africa, Australia, Southeast Asia, Northeast Asia, Eastern China, and India). The dotted line is the trend line.

## 4 Data availability

The visibility-derived AOD at station and grid scales over global land are available at National Tibetan Plateau / Third Pole Environment Data Center, https://doi.org/10.11888/Atmos.tpdc.300822 (Hao et al., 2023).

We provide the station-scale AOD from 1959 to 2021. Due to a small number and sparse meteorological stations prior to 1980, we only provide the gridded AOD from 1980 to 2021. In order to keep consistency in time scale, the time range we describe in this study is from 1980 to 2021. The following is a description to the station and gridded VIS_AOD dataset.

The station-scale AOD files are in 'Station_Daily_AOD_1959_2021.zip'. The station-scale AOD files can be directly opened by a text program (such as Notepad). The details station information is in the file of '0A0A-Station_ In Information.txt'. There are eight columns in each text file, separated by commas and column names are Datetime, TEMP (℃), DEW (℃), RH (%), WS (m/s), SLP (hPa), DRYVIS (km), and VIS_AOD (550nm). The 2-7th column names are temperature (unit: ℃), dew temperature (unit: ℃), relative humility (unit: %), wind speed (unit: m/s), sea level pressure (unit: hPa), and dry visibility (unit: km).

The gridded AOD is in the file of 'Gridded_Monthly_AOD_1980_2021.nc' with a NETCDF4 format. There are three variables: 'VIS_AOD' (AOD derived from visibility), 'W95CI' (the width of the 95% confidence interval), and 'QA_FLAG' (quality flag for VIS_AOD). We classify the quality of VIS_AOD into three levels based on 'W95CI': (1) High quality (QA_FLAG=1):

W95CI<=0.03; (2) Medium quality (QA_FLAG=2), 0.03<W95CI<=0.06; and Low quality (QA_FLAG=3), W95CI>0.06. The more details are in '0A0B-ReadMe.txt'.

## 5 Conclusions

In this study, we employ a machine learning technique to derive AOD for over 5000 land stations worldwide, based on satellite data, visibility, and related meteorological variables. The target is Aqua MODIS AOD. Monthly AOD is interpolated into a 0.5° grid using ordinary kriging with area weighting. The accuracy and performance of the derived AOD are assessed and validated against Terra MODIS AOD as well as AERONET ground-based observations and MRRRA-2 AOD. The gridded AOD is evaluated by Aqua and Terra MODIS AOD and a 95% confidence interval is calculated. We obtain daily AOD (550nm) at 5032 global land stations from 1980 to 2021, as well as monthly gridded AOD. The two datasets complement the shortcomings of AOD data in terms of time scale and spatial coverage. Finally, the spatiotemporal variation in AOD is analyzed for global land, the Southern Hemisphere, the Northern Hemisphere, and 12 regions in the past 42 years. Several key findings have been given in this study as follows.

**1. Modeling and gridding evaluation.** The mean RMSE, MAE, and R of all stations are 0.078, 0.044, and 0.750, respectively. The RMSE of 93% stations is less than 0.11, the MAE of 91% is less than 0.06, and the R of 88% is greater than 0.7, respectively. Compared to Aqua and Terra, the average biases of gridded AOD are 3.3% and 1.9%, and the spatial correlation coefficients are 0.80 and 0.79, with the zonal correlation coefficients of 0.99 and 0.99 and the meridional correlation coefficients of 0.99 and 0.90.

**2. Model validation.** For the daily scale, the R, RMSE and MAE of between VIS_AOD and Aqua AOD is 0.799, 0.079 and 0.044, respectively. The percentage of sample point falling within the EE envelopes is 84.12%. The R between VIS_AOD and Terra AOD is 0.542, with a RMSE of 0.125 and MAE of 0.078. The percentage falling within the EE envelopes is 64.76%. The R between VIS_AOD and AERONET AOD is 0.546, with a RMSE of 0.186 and MAE of 0.099. The percentage falling within the EE envelopes is 57.87%. For the monthly and annual scales, RMSE and MAE show a significant decrease between VIS_AOD and Aqua, Terra, and AERONET AOD, and R and percentages falling within EE show a significant increase.

**3. Error analysis.** The average bias is 0.015 (AOD <0.1), with 83% of data within the EE envelopes. As pollution level increases, the negative mean bias becomes significant and the underestimation increases. There is a negative bias in the low elevation (<=0.5km) with a percentage of 60%-64% falling within the EE envelopes and a positive bias in high elevation (0.5-1.2km) with a percentage of 50%-65% falling within the EE envelopes. The elevation of AERONET's site caused a bias in high elevation. When the elevation difference is negative (the elevation of the meteorological station is lower than that of the AERONET site), there is a significant positive bias. When the difference is positive, the mean bias approaches 0 or is positive. The bias does not change significantly with increasing distance between the meteorological station and AERONET site.

**4. Global land AOD.** The global, NH, and SH AOD values from 1980 to 2021 are 0.161 ± 0.074, 0.158 ± 0.076, and 0.173 ± 0.059, respectively. Trends in AOD for the global, NH, and SH demonstrate a decreasing trend of -0.0026/10a, -0.0018/10a, and -0.0059/10a, respectively (p<0.01).

The seasonal AOD ranking from high to low is JJA>MAM>DJF>SON over the global land and in the NH, while in the SH, it is DJF>JJA>MAM>SON. The largest declining trends are observed in NH summer and SH winter.

**5. Regional AOD.** From 1980 to 2021, regions with high aerosol loadings (AOD > 0.2) were found in West Africa, Northeast Asia, Eastern China, and India. Regions with moderate aerosol loadings (AOD between 0.15 and 0.2) are Eastern North America, Central South America, South Africa, and Southeast Asia. Eastern Europe, Western Europe, Western North America, and Australia are regions with low aerosol loadings (AOD < 0.15). The trends are -0.0041/10a, -0.0021/10a, -0.0009/10a, -0.0021/10a, -0.0100/10a, -0.0008/10a, -0.0096/10a), -0.0081/10a, -0.0003/10a, -0.0000/10a, 0.0071/10a, and 0.0096/10a in Eastern Europe, Western Europe, Western North America, Eastern North America, Central South America, Western Africa, Southern Africa, Australia, Southeast Asia, Northeast Asia, Eastern China, and India, respectively.

# Competing interests

The contact author has declared that none of the authors has any competing interests.

# Acknowledgments

This work is supported by the National Key Research & Development Program of China (2022YFF0801302) and the National Natural Science Foundation of China (41930970). The hourly visibility data are downloaded from https://mesonet.agron.iastate.edu/ASOS. The Aerosol Robotic Network (AERONET) 15-minute aerosol optical depth (AOD) data are downloaded from which can be downloaded from https://aeronet.gsfc.nasa.gov. The MODIS AOD data are downloaded from https://ladsweb.modaps.eosdis.nasa.gov.

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
