# Peer review of "Visibility-derived aerosol optical depth over global land from 1980 to"

_Earth System Science Data, 2023_

## Author Comment (AC1)

**Visibility-derived aerosol optical depth over global land from 1980 to 2021**

**Response to RC1**

We thank the referee for the constructive and helpful comments. We carefully thought about the comments and made corresponding revisions to the manuscript, which have substantially improved the manuscript.

**1. Main modifications to the content:**

(1) Simplified the introduction.

(2) Modified the introduction of meteorological data (section 2.2).

(3) Modified the temporal matching method with AERONET in section 2.5. Added expected error (Eq. 14).

(4) Added error analysis at global, regional, and site scales (sections 3.3.1-3.3).

(5) Added uncertainty analysis with AERONET AOD (section 3.3.4).

(6) Added trend analysis for India (Section 3.6).

(7) Improved dataset files.

**2. Modifications to the chart:**

(1) Modified the Figure 5.

(2) Merged the original Figures 6 and 7 into Figure 6.

(3) Added Table 1, evaluation results for each region.

(4) Added Figure 7, evaluation results at site scale.

(5) Added Figure 8, uncertainty analysis.

(6) Added Figure 12, average AOD for different latitude ranges.

(7) Added India in Figures 1 and 13-15.

***Comment 1. Is the article itself appropriate to support the publication of a data set?***

*The data set is not a novel idea, and its premise is rooted with errs in assumptions. The article does not indicate how it improves over data assimilation techniques already implemented by various numerical weather prediction and reanalysis systems.*

*Vertical resolution of the aerosol is needed to adequately link the surface visibility to the total column. The visibility from AOD technique suffers similarly from the same fundamental problem investigated for more than two decades to accurately determine PM2.5 from space in the presence of aerosols above the surface boundary layer. This data set continues to have the same drawbacks as techniques used in the 2000s and the results show the Vis AOD data set does not address this key aspect with the stated machine learning methodology. When providing this data set, this major drawback must be clearly stated to the user when considering using these data. The use of vertical aerosol measurements may improve this data set and make it more useable.*

**Response 1.**

● We agree with the referee on the difference between surface visibility and aerosol

optical depth, which is discussed in the 9th paragraph of the introduction. To address this issue, we included the boundary layer height into our method, as the most aerosols locate in the boundary layer. The evaluations in this study demonstrate that this method can provides reliable dataset in depicting annual and long-term trend. As we know, this study provides the first global AOD over land from 1980 to 2021 based on visibility.

- The primary purpose of the study is to provide the global AOD over land for the time period before the EOS satellite era (i.e., before 2000) and analyze trends in global land and regions, which provide a unique data for broad user. We did not intend to improve data assimilation techniques implemented by various numerical weather prediction and reanalysis systems.

- We agree with that the profile of aerosols is the best way to estimate AOD based on surface visibility, however, the reliable dataset on aerosol profile is not globally available. We have discussed the possible errors in section 3.3.4.

**Comment 2. Is the data set significant – unique, useful, and complete?**

*The data is available via HTTP download with a short delay but data is not bundled optimally. The data set provides a zip file. Inside the zip file, it provides "Gridded_Monthly_AOD_1980_2021" and another zip file "Station_Daily_AOD_1980_2021.zip."*

**Comment 2.1**

*Inside the "Gridded_Monthly_AOD_1980_2021" directory, a file is provided in netCDF4 format with compatibility for HDF5. The netCDF file has four variables, time "T", Latitude "Lat", Longitude "Lon", and visibility to AOD "VIS_AOD". The netCDF file lacks various metadata such as standard_name, _FillValue, valid_range, long_name, and coordinates. The shape "VIS_AOD" is 504 (Time), 721 (Longitude), and 291 (Latitude). The latitude dimension has an unusual shape since the manuscript defines the latitude range from -60N (or 60S) to 85N. This spatial extent is not provided in the metadata or the data landing page for TPDC.*

**Response 2.1**

- Thank you for your suggestion. We follow the referee suggestions and re-uploaded the dataset. (1) The variables in the netCDF file have been modified to "time", "longitude", "latitude", and "VIS-AOD". (2) We have added the missing metadata data in the netCDF file. (3) The shape (-60°N to 85°N) of latitude dimension is based on the distribution of meteorological stations. (4) We have added the spatial extent in the metadata and the data landing page in TPDC, which is waiting for the data center manager's approval.

**Comment 2.2**

*Inside Station_Daily_AOD_1980_2021.zip, a directory called "Station_Daily_AOD_1980_2021" has ASCII text files that are named for the ASOS station with Vis AOD data in which the "VIS_AOD" column does not indicate the wavelength (i.e., 550nm). A separate "0A0A-Station_Information.txt" file provides the ASOS station names and associated longitude, latitude, and elevation (note that data units are not provided) in this file.*

**Response 2.2**

- Thank you for your suggestion. We have added data units in the file of station information and uploaded to the data center.

***Comment 2.3***

*The manuscript states at line 462, "The more sample data input, the better the model performs." Typically, 80% should be used for training and 20% for testing. Using all of the input data leads to weaker determination of the performance of the model.*

**Response 2.3**

- Thank you for your comment. We agree with referee and have attempted to use 80% for training and 20% for testing. The reasons for not using this method are:
  (1) Figure 3 shows that models perform better using more sample data.
  (2) More importantly, we have two independent datasets for evaluation (Terra and AERONET AOD).

***Comment 2.4***

*Figure 6 shows VIS_AOD has wide variability from zero to near ~0.7 when compared to all three input sources. These differences could lead to very large variations in daily climatology analysis at specific locations. In Figure 7, the correlation improves by increasing the temporal average window which acts to smooth errors in the model. However, in Figure 7, the bottom right panel showing Annual AERONET (labelled incorrectly as TERRA), still indicates a much weaker correlation as errors in modelling and systematic biases and uncertainties with MODIS AOD input products are still evident.*

**Response 2.4**

Thanks for your comments. The type error has been corrected. According the comments from referee #1 and #2, we made the following revision:

- We have made modifications in original Figure 6 and Figure 7 (Figure 6 in the revised version). 15-minute AERONET AOD (550nm) is used to validate and the expected error ($\pm$ (0.05+0.15 * $AOD_{AERONET}$)) is used to evaluate VIS_AOD, which show higher correlation coefficient in daily scale.

- We added more discussion. In Section 3.3, the error analysis results show that VIS_AOD is underestimated in heavy pollution. It has led to significant changes in daily scale climate analysis at specific locations in section 3.3.3 Validation at a site scale. The discussion on averaging over time scales (Schutgens, et al., 2017) was also added. The error analysis and limitations for VIS_AOD are discussed in section 3.3.4 (Eck et al., 2023; Levy et al., 2013; Levy et al., 2018; Li et al., 2020; Wei, et al., 2019a; Wei, et al, 2020; Zhang et al., 2020).

***Comment 3. Is the data set itself of high quality?***

***Comment 3.1***

*Checking Vis AOD with AERONET AOD daily averages shows an example of deviations that often occur when elevated aerosol layers impact both AERONET measurements and MODIS retrievals compared to the surface visibility.*

*For more background on this case, see the following article (Eck et al., 2023): https://doi.org/10.1016/j.atmosenv.2023.119798*

*Aerosol transport cases from biomass burning smoke and dust affect many sites around the globe. These episodic smoke transport events are increasing as drought severity due to global warming continues to promote drier conditions in vegetated and desert regions. The world's deserts such as Saharan, Gobi, and Thar, desert dust continue to be key sources of aerosols lofted above the boundary layer and transported 100s of kms. Other elevated aerosols include volcanic ash and gas-to-aerosol conversion of sulfur dioxide to sulfates in the upper and lower stratosphere such as the recent Hunga Tonga volcanic eruption.*

**Response 3.1**

- Thank you for using our data and providing an example of the bias between daily VIS_AOD and AERONET AOD during aerosol transport events. We agree with your point and have carefully read the background information of the case. The difference between surface visibility and column aerosols is a key factor causing this deviation. We discussed it in section 3.3.4 and cited the reference (Eck et al., 2023). Figure 5 show in the revised version show that VIS_AOD has a comparable accuracy of TERRA AOD and AQUA AOD at daily, monthly and yearly time scales.

*Comment 3.2*

The manuscript indicates ASOS data (https://www.weather.gov/asos/faq.html#12) are retrieved from the Iowa State University (https://mesonet.agron.iastate.edu/ASOS/).
This Iowa State University web page indicates the following information on the U.S. ASOS network which does not refer to the international weather observation stations that are not maintained by the U.S. NWS, FAA, or DOD. While the Iowa State University search tool indicates "ASOS" for the international locations, it is incorrect to assume that they are managed by the U.S. Each international network has their own method for monitoring surface weather using different instrumentation and methods; however, some stations such as international airports need to meet internationally mandated regulations (e.g., WMO: https://community.wmo.int/en/implementation-areas-aeronautical-meteorology-programme), but these may not apply to other stations within the country. Therefore, the comprehensiveness of surface weather data at the international locations may not be as robust quality or measurement accuracy as those collected at international locations outside international airports. Assertion that the global network of surface meteorological measurements is managed by the U.S. ASOS network is misleading.

**Response 3.2**

- Thank you for your correction. We agree with your opinion. The introduction of meteorological data has been modified in section 2.2.

*Comment 3.3*

*AERONET sites often are not collocated with ASOS measurements. The representativeness of the AERONET site compared to the ASOS site needs to be considered. Did the Authors consider the elevation of the AERONET site with respect to the ASOS location? Oftentimes, AERONET sites can be elevated and further above the ASOS location in urban, forested, mountainous or marine locations in which they*

*are placed on towers or buildings 10s of meters high above ground. The difference in elevation can have a significant effect on the visibility relationship to the AOD measurement. Therefore, the measurement of the aerosols may be less at the measurement altitude of AERONET site compared to the elevation of the ASOS location.*

**Response 3.3**

- Thank you for your suggestion. We have added error analysis and uncertainty analysis for VIS_AOD in section 3.3.4.2-3.3.4.4 and a new figure was added (Figure 8 in the revised version) to show: (1) Elevation of AERONET site and bias (Figure 8 (b)), (2) Elevation difference between AERONET site and meteorological station and bias (Figure 8 (c)), (3) Distance between AERONET and meteorological station and bias (Figure 8 (d)).

*Comment 4. Is the data set publication as submitted of high quality?*

*Comment 4.1*

*At lines, 511-513, the statement: "However, the AERONET AOD results are slightly inferior to those of Aqua and Terra AOD, which could be caused by the representativeness of the AERONET station spatial coverage and measurement error (Holben et al., 1998)" is interpreted out of context. AERONET AOD are superior in the determination of AOD. The determination of the spatiotemporal representativeness is within the purview of the methodology utilized to perform the matchup and the justification for such a methodology to improve the spatiotemporal representativeness. Therefore, the fact that MODIS AOD follows closely to the Vis AOD suggests that this input is significantly weighted to it. This is obvious from the variation in the stated correlation coefficients between the model and the input data sets where AERONET is much weaker, and MODIS is much stronger.*

**Response 4.1**

- Thanks for your comments. We follow the referee's suggestion to revise the matching method between AERONET and visibility derived AOD for comparison. We used a 15-minute AERONT AOD for spatiotemporal matching and validation. Figure 5 show in the revised version show that VIS_AOD has a comparable accuracy of TERRA AOD and AQUA AOD at daily, monthly and yearly time scales. We added new discussion on the differences between VIS_AOD and MODIS AOD and AERONET AOD in section 3.3.4.

*Comment 4.2*

*Lines 513 – 515 states "Nevertheless, the results indicate the high reliability and strong predicted capability of the model, and the visibility-derived AOD can be used for aerosol climatology." A major issue with this statement is that areas of the world are affected by transported aerosol above the boundary.*

**Response 4.2**

- Thank you for your correction. We agree with your opinion. We have made the modifications and discussed in section 3.3.4.

*Comment 4.3*

*What is the uncertainty of the boundary layer height for the ERA5 reanalysis? Please state.*

**Response 4.3**

- Thank you for the suggestion. We have added the uncertainty of the boundary layer height for the ERA5 reanalysis in section 2.3.

*Comment 4.4*

*The netCDF file lacks various metadata such as standard_name, _FillValue, valid_range, long_name, and coordinates.*

**Response 4.4**

- Thank you for the suggestion. We have added the lack metadata in the netCDF file and reuploaded the dataset to the data center.

*Comment 4.5*

*Figure 2 lists Aqua MODIS twice and does not indicate Terra MODIS.*

**Response 4.5**

- Thank you for the correction. We have made an adjustment in Figure 2.

*Comment 4.6*

*Figure 7 shows the lower right panel the same x-axis title as the middle right panel.*

**Response 4.6**

- Thank you for the correction. We have removed Figure 7 and replaced it with Figure 6.

---

## Author Comment (AC2)

**Visibility-derived aerosol optical depth over global land from 1980 to 2021**

**Response to RC2**

We thank the referee for the constructive and helpful comments. We carefully thought about the comments and made corresponding revisions to the manuscript, which have substantially improved the manuscript.

**1. Main modifications to the content:**

(1) Simplified the introduction.

(2) Modified the introduction of meteorological data (section 2.2).

(3) Modified the temporal matching method with AERONET in section 2.5. Added expected error (Eq. 14).

(4) Added error analysis at global, regional, and site scales (sections 3.3.1-3.3).

(5) Added uncertainty analysis with AERONET AOD (section 3.3.4).

(6) Added trend analysis for India (Section 3.6).

(7) Improved dataset files.

**2. Modifications to the chart:**

(1) Modified the Figure 5.

(2) Merged the original Figures 6 and 7 into Figure 6.

(3) Added Table 1, evaluation results for each region.

(4) Added Figure 7, evaluation results at site scale.

(5) Added Figure 8, uncertainty analysis.

(6) Added Figure 12, average AOD for different latitude ranges.

(7) Added India in Figures 1 and 13-15.

*This manuscript describes the dataset of global land AOD from 1980 to 2021 derived from visibility data (VIS_AOD) using the machine learning method. The Aqua MODIS AOD data were used as the training dataset and the resulted VIS_AOD were evaluated with AOD from Terra MODIS and AERONET, and the trends of annual and seasonal mean VIS_AOD over several regions were assessed.*

*The most significant value of the dataset is that it provides the global AOD over land for the time before the EOS satellite era (i.e., before 2000). With that, I recommend the manuscript to be published on ESSD. However, I do have several comments that should be addressed in the revision before it is accepted for publication.*

*Major comments:*

*Comment 1: Introduction: The introduction section is unnecessarily too long. I read paragraphs after paragraphs and still did not get what this paper was about until near the end of page 5. I suggest significantly shorten the introduction to briefly introduce aerosols, tell the readers why AOD is important, what the limitation of available satellite and AERONET data records are, and what this paper is about.*

**Response 1:** Thank you for your suggestion. We removed content unrelated to AOD. We have mainly simplified the first, fourth, and eighth paragraphs in the introduction.

**Comment 2:** *Limitation of the VIS_AOD: fundamentally, visibility observations are at the surface, but AOD is a column integrated quantity. Considering PBL height is necessary in converting the surface quantity to column AOD but it is not sufficient, because aerosols are frequently located above the PBL, especially in the cases of large fires and transported plumes. In addition, different aerosol species have different optical properties, and the aerosol composition in the vertical column could be quite different from that near the surface. It is thus necessary to discuss the limitations and uncertainties or errors associated with the VIS_AOD products.*

**Response 2:** Thank you for your suggestion. We used 15-minute AERONETAOD (550nm) and re-evaluated VIS_AOD using expected error ($\pm(0.05+0.15*AOD)$). We have discussed the limitations in and the uncertainties in section 3.3.4, and errors in section 3.3.1-3.3.3.

**Comment 3:** *Validation: AERONET AOD is considered as a "ground truth" because of the direct measurements and globally unified, rigorous standard calibration. AERONET AOD has been used extensively for satellite retrieval validations, including the Terra and Aqua MODIS products. The correlation coefficients of VIS_AOD vs. AERONET AOD is only 0.51 for coincidental data, which is much lower than that of MODIS AOD vs. AERONET (R=0.86, Levy et al., 2013 for MODIS C6 AOD products). The VIS_AOD quality and uncertainties again need to be assessed with the AERONET AOD.*

**Response 3:** Thank you for your suggestion. We matched 15-minute AERONET AOD at 550nm and re-evaluated VIS_AOD using expected error ($\pm(0.05+0.15*AOD)$). The VIS_AOD quality and uncertainties have been assessed with the AERONET AOD in section 3.3.3 and 3.3.4. In this study, we used all the available globally AERONET AOD data which cover a period of 20 years.

**Comment 4:** *Analysis of AOD variations: It is poorly done. There are many unsubstantiated claims, inconsistent explanations, etc. Section 3.5 requires major revision. See my specific comments below.*

**Response 4:** Thank you for your suggestion. We have made an adjustment and improvement for the analysis of AOD variations in section 3.5 and 3.6.

*Specific comments:*

**SC1:** *Line 39: "aerosol particles are primarily discharged from the Earth's surface" – this is not correct. An overwhelming majority of inorganic aerosols (e.g., sulfate, nitrate) are form in the atmosphere via photochemical reactions of their gaseous precursors.*

**SC1 Response:** Thank you for your correction. We have made modifications. Aerosol particles are directly emitted into the atmosphere or formed through gas-particle transformation.

**SC2:** *Line 40-43, cited references: There are many previous publications for the related topic, yet only single citation is listed as if that is the only reference or the original one. This is not appropriate. At least you should add "e.g." in front of the references and add a few more.*

**SC2 Response:** Thank you for your suggestion. We have added references.

*SC3: Line 67-68: What are the major deficiencies? How they contribute to the uncertainties of climate forcing?*

**SC3 Response:** Thank you for your suggestion. We have made modifications. The uncertainties are caused by the deficiencies of the global descriptions of aerosol optical properties (such as scattering and absorption) and microphysical properties (such as size and component), and the impact on cloud and precipitation, further affecting the estimation of aerosol radiative forcing.

*SC4: Line 144: "inherent limitation of long temporal coverage" - How long is the solar radiation data that can be used to infer AOD? From the reference sited in line 140, the first one started at least in 1978 or before.*

**SC4 Response:** This part is not related to visibility-based AOD, and we have deleted it.

*SC5: Line 148-151: How do the observations of extinction, water vapor, and gas molecules at the surface help you in this work?*

**SC5 Response:** Thank you for your suggestion. We have made modifications. We reduced the sensitivity of particulate matter to humidity by calculating dry visibility. The optical depth of gas molecules is a quantity related to position and elevation, and is considered a constant (Li et al. ,2020).

*SC6: Line 179-180: The challenges listed in this paragraph (lines 171-178) exists everywhere on local, regional, and global scales. Why does the AOD can be done from visibility data regionally less challenging than globally? You are doing global anyway in this study.*

**SC6 Response:** Thank you for your suggestion. We have adjusted this sentence.

*SC7: Figure 1 on page 6: Why India is not included in the trend analysis? It is an important region that has undergone rapid changes.*

**SC7 Response:** Thank you for your suggestion. We have added India (region 12 in Figure 1) and analyzed the trend in India in Figures 13-15.

*SC8: Line 217-218, Figure 1 caption: The order of these two regions is reversed in the figure caption. Northeast Asia is labeled 11 and Eastern China 12 on the figure.*

**SC8 Response:** Thank you for your correction. We have made an adjustment.

*SC9: Line 229, eq.1: I wonder what advantage is to harmonic mean instead of other ways to calculate the mean.*

**SC9 Response:** We have added the advantages of harmonic mean in section 2.2:
(1) The extinction coefficient is directly proportional to the reciprocal of visibility, so the result of harmonic averaging is more reasonable.
(2) Harmonic averaging can capture the decrease in visibility faster than arithmetic averaging. For example, data with 6 1-minute intervals, 10,10, 1,1,1,1 km. The arithmetic average result is 4km, and the harmonic average result is 30/21km.
(3) Daily representativeness. Combining (1) and (2), daily averages is more representative on the daily scale.

**SC10:** *Line 248-251: the sentence "Because…are adopted" sounds strange: Because three variables (RH, pressure, wind speed) are related to aerosol properties, five variables are adopted. Explain why you also adopt TMP and WS.*

**SC10 Response:** Thank you for your suggestion. We have made modifications in section 2.2. In addition to hourly visibility (VIS), some automatically observed variables closely related to aerosol properties were selected, including relative humidity (RH), dew point temperature (DT), temperature (TMP), wind speed (WS) and sea-level pressure (SLP). Temperature affects atmospheric stability and the rate of secondary particle formation, and humidity influences the size and hygroscopic growth, and wind speed and pressure significantly impact the transport and deposition.

**SC11:** *Line 257-259: What happens when RH is out of that range (30-90%)?*

**SC11 Response:** We have given an explanation in the manuscript, as follow: When the relative humidity is less than 30%, the dilution effect of aerosols is very low or even negligible. When the relative humidity is greater than 90%, research shows that it is impossible to distinguish whether it is fog or haze, or both exist at the same time, and even precipitation.

**SC12:** *Line 262-263: Please give reasons for using different methods calculating means for different variables.*

**SC12 Response:** We have provided an explanation in SC9 Response.

**SC13:** *Line 279: Explain why these three variables are needed.*

**SC13 Response:** We have provided an explanation in section 2.3.

**SC14:** *Line 355-359: The language used in these lines are unclear and confusing. Does "good weather conditions" mean low AOD and clear sky? Does "AOD values are concentrated around the average value" mean AOD variability is small? Does "bad weather" mean heavy pollution events? What does "data imbalance" mean? Balance with what? How large is AOD that can be described as "large AOD"?*

**SC14 Response:** We have made modifications in section2.6.2. When it is clear, the AOD value is small, the variability of AOD is small (AOD <0.5), and the data is concentrated near the mean value. When heavy pollution, the AOD value is large (AOD >0.5). Compared to clear sky, the AOD sequence will show "abnormal" large values with low frequency, which is the imbalance of AOD data. The processing of imbalanced data. (1) AOD sequences are classified into three types based on percentile (0-1%, 2% -98%, 99%). (2) When the mean of the third type of AOD is greater than 5 times the standard bias of the second type, it is considered an imbalanced sequence. These data, with a total amount less than 5% of the sample, are imbalanced data. (3) Then synthetic samples are generated with the upper limit 10% of the samples.

**SC15:** *Line 432-433: Explain the symbols in Eq. 14-16.*

**SC15 Response:** We have made supplement.

**SC16:** *Section 3.3: How is the daily value obtained? Do you match the time and location of observations between AERONET and MODIS? How is your evaluation of MODIS vs. AERONET compared to many published MODIS validation papers?*

**SC17 Response:** We have made modification in section 2.5. In the previous comparative analysis, time and location were not considered when matching AERONET with satellite AOD. At present, the matching with AERONET AOD (550nm) is at least two times within 1 hour (± 30 minutes) of satellite transit time.

*SC17: Line 492: when comparing with satellite daily, monthly, and yearly data, do you match the time and location to do the means at these time scales?*

**SC18 Response:** We have made supplements. We matched the time and location with satellite at daily, monthly, and yearly scales.

*SC19: Line 494-501, validation of VIS_AOD with MODIS and AERONET: Because you use Aqua AOD to train the visibility-based ML model, the comparison between VIS_AOD and Aqua AOD is not an independent validation. Please clarify.*

**SC19 Response:** Thank you for your correction. We have clarified that Aqua AOD is not an independent validation in section 3.3.

*SC20: Line 510-513: AERONET AOD is considered "ground truth" and is more accurate than MODIS because it is a direct measurement, not a retrieved product. All satellite AOD products have been evaluated with AERONET for their quality. Maybe you should train your model with AERONET AOD instead of Aqua MODIS AOD. In fact, AERONET AOD should be the standard data product to evaluate the model error.*

**SC20 Response:** Thank you for your suggestion. We have attempted AERONET AOD as the target value for the model. However, the AERONET site is sparse and the observation period is inconsistent, making it insufficient for gridding. Therefore, we chose MODIS AOD to train the model and AERONET AOD as standard data to evaluate model errors.

*SC21: Y-axis title on Figure 7: It should be "Annual AERONET_AOD", not "Annual Terra_AOD" for the lower right panel.*

**SC21 Response:** Thank you for your correction. We have removed Figure 7 and replaced it with Figure 6.

*SC22: Line 528: Clarify that the zonal and meridional AOD are over land only.*

**SC22 Response:** Thank you for your correction. We have clarified that the zonal and meridional AOD are over land only.

*SC23: Line 532: Like mentioned above, the evaluation between VIS_AOD and Aqua MODIS AOD is not meaningful and not an independent evaluation.*

**SC23 Response:** We have made modification. We have indicated that Aqua AOD is not an independent validation, while Terra and AERONET are independent validation.

*SC24: Line 536: How to define "highly similar": need to be quantitative and avoid subjective phrases.*

**SC24 Response:** Thank you for your correction. We have made modifications. The R between Aqua and Terra AOD are highly similar, with an R of is 0.980.

*SC25: Line 553: "good agreement" – this is another subjective phrase. The R between VIS_AOD and AERONET AOD (remember, AERONET AOD is the actual*

*measurement, not a retrieval product) is 0.624, meaning it captures 39% of observed AOD. Is it good enough?*

**SC25 Response:** Thank you for your correction. We have deleted this sentence.

*SC26: Line 561-562: "High AOD values occur in the NH…", which contradicts with the numbers in line 559 (mean AOD of NH=0.158 and SH=0.173)!*

**SC26 Response:** Thank you for your suggestion. We have made modifications. There are many regions with high AOD values over land in NH.

*SC27: Line 566-568: Figure 8 shows that lower AOD value is not just at 25S, but all latitude from 25S to 60S. In any case, Figure 8 does not support the higher SH AOD than NH AOD.*

**SC27 Response:** Thank you for your suggestion. We have made modifications. Lower AOD regions of the SH are from -25°N to -60°N.

*SC28: Line 590-592: Why should industrial activities intensify in JJA? Is there any data support that? Reasons for higher AOD in the summertime has a lot to do with the higher RH to promote aerosol growth, and more active photochemical production of aerosols from their precursor gases.*

**SC28 Response:** Thank you for your correction. We have made modifications. Under higher relative humidity in JJA, it promotes the growth of hygroscopic particle and the photochemical reaction of aerosol precursors, resulting in high aerosol loading.

*SC29: Line 591-592: SON is biomass burning season in South America and southern Africa, and the high AOD there and then is not due to "intensification of industrial activities"!*

**SC29 Response:** Thank you for your correction. We have made modifications. The occurrence of high AOD values is highly associated with the growth of hygroscopic particle and the photochemical reaction of aerosol precursors under higher relative humidity in Asia (JJA) (Remer et al., 2008) and Europe such as Russia (JJA), and biomass burning in South America (SON), Southern Africa (SON), and Indonesia (SON).

*SC30: Line 593-594: "the increased dust emission in Middle East region related to the transport of dust from the Sahara region": This sentence does not make any sense. How can transport of dust from Sahara cause the increased dust emission in Middle East?*

**SC30 Response:** Thank you for your correction. We have made modifications. We replaced the Middle East with India

*SC31: Line 595-596: Monsoon systems are most active in summer, not autumn!*

**SC31 Response:** Thank you for your correction. We have made modifications. This may be related to the weakening of the monsoon.

*SC32: Line 601-603: Why does AOD in SH decrease much faster than that in in NH?*

**SC32 Response:** Thank you for your suggestion. We have given an explanation. There is a decrease in the frequency of sandstorms and wildfires and an increase in precipitation, such as in Australia.

*SC33: Line 603-604, MODIS trends: You should compare your trends for the same period 2003-2020 with the MODIS trends over land only.*

**SC33 Response:** Thank you for your suggestion. We have made the modifications.

*SC34: Line 607-608: "our study has the same downward signal as that in previous studies": This is not true from the trend values presented just a few lines above. Your trends are over land only and they are in opposite directions to MODIS and SeaWiFS!*

**SC35 Response:** Thank you for your correction. We have made the modifications.

*SC35: Line 614-616: Are these two sentences or one? It seems the period should be replaced with a comma on line 615.*

**SC36 Response:** We have made modification. It is one sentence.

*SC37: Figure 10: (1) in the caption, symbols are different from what depicted in the legends on the first right panel. Is triangles representing NH (in caption) or SH (on legends)? If the yellowish lines with triangles are for NH, why the global means are much closer to SH, given that NH has much larger land surface? If they are for SH, why AOD in SH is much higher than that in the NH for all seasons and all times except in MAM after 2000? There is a lot of explanations to do.*

**SC37 Response:** (1) The line types and markings in Figure 11 have been corrected. (2) The triangular yellow line represents SH, and the square red line represents NH. Figure 12 illustrates the multi-year average AOD in different latitude ranges for land, the NH, and the SH from 1980 to 2021.

*SC38: Line 649: Again, I wonder why India is excluded.*

**SC39 Response:** We have added **India** into this study.

*SC40: Line 655-657: The large volcanic eruptions from El Chichon and Mt. Pinatubo do not appear in most regions, which are not correct. Is this because your training data are after 2002 that do not have any knowledge of major volcanic eruptions? Does this illustrate the limitation of visibility-based estimates of column AOD?*

**SC40 Response:** Thank you for your suggestion. Volcanic eruptions also have a significant impact on tropospheric aerosols. The presence of tropospheric aerosols will affect visibility near the ground. Although the model does not have data from before 2002, visibility had the records and captured the characteristics of historical aerosols. We think that it is not a limitation of visibility-based estimation, but rather its advantage.

*SC41: Line 658: It is AOD, not loading, that is shown in Figure 11.*

**SC41 Response:** Thank you for your suggestion. It has been modified to 'AOD level'.

*SC42: Line 688: Why are these three numbers shown 4 significant digits after the decimal point, but all other numbers are with just 3?*

**SC42 Response:** Thank you for your correction. We have ensured consistency in significant digits after the decimal point.

*SC43: Line 703-804: Do you get the sign correctly or consistently? How can a negative number (-0.01/10a) be greater than a positive number (0.009/10a)?*

**SC43 Response:** Thank you for your correction. It is a writing error and the negative sign was missed. It has been corrected to -0.009/10a.

*SC44: Line 715: JJA is NOT the biomass burning season in West Africa.*

**SC44 Response:** Thank you for your correction. We have made modifications.

*SC45: Line 729: Wildfire is very seasonal. Does your seasonal trend support the conclusion of BC and OC decreasing?*

**SC45 Response:** Thank you for your corrections. We have made modifications. It does not support the decreasing in BC and OC, and it is related to dust, biomass burning, and forest cover area.

*SC46: Line 744: "Natural emissions were predominant in 1992 and 1997": What is the base of such statement? Figure 12 and 13 hardly show any enhancement of AOD in these years.*

**SC46 Response:** Thank you for your correction. We have deleted this sentence.

*SC47: Line 749-750: Why should PBL height affect AOD, which is a column quantity? This is another unsubstantiated claim.*

**SC47 Response:** Thank you for your correction. We have made modifications, 'the reason for the high AOD in winter may be related to the transportation.'

*SC48: Line 757-758: What does "high level of volatility" of AOD mean?*

**SC48 Response:** We have made modifications, 'larger positive anomaly accompanied by oscillations.'

*SC49: Line 773: "201" should be 2021.*

**SC49 Response:** Thank you for your correction. We have made modifications.

*SC50: Figure 12: Discussions of the differences between VIS_AOD and MODIS_AOD should be explained.*

**SC50 Response:** Thank you for your suggestion. We have made an explanation for the differences in section 3.3.4.

---

## Author Response (AR3)

**Visibility-derived aerosol optical depth over global land from 1980 to 2021**

**Response to Anonymous Referee #1 and #3 (AR1 and AR3)**

We thank the referee for the constructive and helpful comments. We carefully thought about the comments and made corresponding revisions to the manuscript and the datasets, which have substantially improved the manuscript and the datasets.

1. Improved the underlying data products.

■ Added other meteorological variables into the station product.

■ Added 95% CI and quality flags into the gridded product.

■ Discussed the uncertainty for gridded product in section 3.4.1.

■ Added a description in section 4.

2. Discussed the uncertainty in regions with sparse stations, such as high-latitude and desert, in section 3.3.3 and 3.4.1.

3. Added new figures to compare VIS_AOD with AERONET AOD (Figure 6) and MERRA-2 AOD (Figure 7) before/after 2000.

**Response to Anonymous Referee #1 (AR1)**

***General Comments:***

*In general, the document has improved from the first version, but some questions remain regarding usefulness of this data set.*

***Data Access and Interpretation:***

***GC 1.*** *The data can be accessed via the National Tibetan Plateau Data Center. The data can be downloaded via FTP using the provided credentials on the web site or they can be downloaded without login but after a wait of up to several minutes. The files provided include netCDF data file with monthly gridded VIS_AOD AOD 550nm values as well as a metafile description. Further, a text file includes the weather station locations where visibility measurement was performed and provides only the "VIS_AOD" result and not the station visibility or any other meteorological information that could be useful in interpreting these data.*

**Response for GC 1:**

● We have added other meteorological variables (temperature, dew point temperature, relative humidity, wind speed, sea level pressure, dry visibility) and the corresponding units into the station VIS_AOD product.

***GC 2.*** *The netCDF file "VIS_AOD" variable needed to be transposed (rows and columns swapped) to enable plotting with Python's matplotlib module.*

**Response for GC 2:**

● We have transposed (rows and columns swapped) the "VIS_AOD" variable and supplemented the "W95CI" variable (the uncertainty with the 95% confidence interval) and the "QA_FLAG" variable (the quality flag of VIS_AOD).

***GC 3.*** During the northern hemisphere winter, landmasses at high latitude are in

darkness for significant periods. How are the visibility data utilized during these periods?

**Response for GC 3:**

● We have excluded visibility records under "blowing snow" weather at high latitude (Husar et al., 2000). In the darkness, visibility can be considered as nighttime visibility and did not perform any other special processing.

***GC 4. In data spare regions such as over Greenland and northern and central Asia, how well does this method perform. It seems that these regions should be excluded from the analysis due to low data availability.***

**Response for GC 4:**

● We have quantified the uncertainty and provided a 95% confidence interval based on kriging variance, as detailed in section 3.4.1. We also have supplemented the uncertainty and the quality flag of VIS_AOD into the gridded product.

***GC 5. When viewing the gridded monthly data, it was obvious data anomalies are present. For example, in northwestern Europe in January 2009, the monthly average over this region showed values between 0.5 and 0.6. These monthly values are very high for a monthly average in this region and generally rare in this region even during periods of high aerosol. When examining visible satellite images for this period and region, it is evident that clouds dominated the view and when they parted it could be determined that snowfall had encompassed the landscape. What controls are in place to restrict visibility measurements in the presence of snow?***

**Response for GC 5:**

● We have excluded low visibility records caused by "blowing snow" at high latitude during data preprocessing (Husar et al., 2000).

***GC 6. For the same month (January 2009), the area of the Indo-Gangetic plain in Northern India shows very low VIS_AOD (generally 0.1-0.2) however this area was highly polluted during this period and monthly average values from satellite and AERONET exceeded 0.5. Missing this magnitude of AOD over a large region gives concerns to the applicability of this approach of converting surface-based visibility to column AOD.***

**Response for GC 6:**

● We have supplemented and discussed the uncertainty of VIS_AOD in high latitude and regions in sections 3.3.3 and 3.4.1.

***GC 7. Given the two cases above, the reviewer examined some other months and found similar issues. How should the user interpret these types of issues? Figure 6 clearly shows these anomalies exist. It should be specified that these anomalies will likely translate to the period 1980-2000 in which this data set is meant to be applied according to the Authors. In this regard, have the authors made an assessment on how their technique compares to data assimilation model such as ERA or MERRA-2? Perhaps the few measurements from AERONET before 2000 could be used to assess the performance of the VIS_AOD with respect to AOD provided by ERA and/or MERRA-2.***

**Response for GC 7:**

- We have added a comparison with AERONET AOD (Figure 6 d, f, i) and MERRA-2 AOD (Figure 7) before/after 2000 in section 3.3.1.

***Technical Manuscript Comments:***

***TC 1.*** *Line 121: What does "1126 ground stations" mean? The sentence is referencing Holben et al which is AERONET. In 2002, only about 150-200 sites existed, not 1126 sites.*

**Response for TC 1:**

- Thank you for your correction. We have counted the number of AERONET sites with 15-minute observations of level 3 in 2002, totaling 152. We have made modifications in the manuscript.

***TC 2.*** *Line 426: tau_target variable is defined as "Aqua MODIS" in Figure 2 and not AERONET/Terra MODIS as stated here.*

**Response for TC 2:**

- To avoid confusion, we have replaced "target" with "true".

***TC 3.*** *Line 429: Validation should be used from AERONET and not Terra MODIS. Terra MODIS is not a validation data set. For inputs, using MODIS from Terra or Aqua will pass the biases associated with them into the model. Why not use only AERONET? Also, training the model with the Aqua MODIS (afternoon overpass) and validating with the Terra MODIS (morning overpass) may have some implications in different meteorological environments (e.g., more clouds in the afternoon).*

**Response for TC 3:**

- We use ground-based AERONET AOD and spaceborne Terra MODIS AOD as independent validation datasets. AERONET provides "true" AOD. Despite changes in the meteorological environment, Terra AOD is also meaningful in long-term data (such as on the monthly and yearly scales) validation and provides a large number of comparable samples over global land.

***TC 4.*** *Line 515-516: Can a plot be added to show the comparison with AERONET prior to 2000?*

**Response for TC 4:**

- We have added the comparison with AERONET before 2000 in Figure 6.

***TC 5.*** *Line 518: VIS_AOD shows a large range of points for the daily average especially around 1 +/-1 AOD. This is a huge range in terms of variation of the AOD. More explanation and investigation into the variation is needed to better understand these variations. Even the monthly average plots have a wide range yet more narrow due to averaging of the errors.*

**Response for TC 5:**

- We have provided explanation and investigation in section 3.3.3.

***TC 6.*** *Line 556: change "word" to "world"*

**Response for TC 6:**

- We have made the modifications.

***TC 7.*** *Line 573: change "AERONT" to "AERONET"*

**Response for TC 7:**

- We have made the modifications.

***TC 8.*** *Line 571-572- Why do these regions have high RMSE?*

**Response for TC 8:**

- We have added explanations in section 3.3.3.

***TC 9.*** *Line 578: "Except for Asia" is disconnected from the previous sentence.*

**Response for TC 9:**

- We have made the modifications.

***TC 10.*** *Line 610-612: If emissions and terrain effects are persistent, then these would be systematic then though?*

**Response for TC 10:**

- We have made modifications to this sentence. Averaging over time scale can reduce representation errors effectively, and emission sources and orography can increase representation errors (Schutgens et al., 2017).

***TC 11.*** *Line 799: Please check header formatting.*

**Response for TC 11:**

- We have made the modifications.

**Response to Anonymous Referee #3 (AR3)**

*Comments:*

*1. Vis-derived AOD does not appear to be superior to satellite retrievals. Just as the authors claim that this method is essential for obtaining long-term AOD before the satellite era, I'm very curious if we can have Vis-derived AOD since the 1950s? if not, could you please say a few words why?*

- **Response:** We have achieved visibility-derived AOD since 1959 and updated it into the station VIS_AOD product.

*2. I am a bit worried about the interpolation of AOD, as there are very few stations in many regions, e.g. the Far East, so caution should be taken with the gridded AOD. In fact, I am not comfortable with the gridded product because in my opinion, large uncertainties are not free from the interpolation.*

- **Response:** We have quantified the uncertainty and provided a 95% confidence interval based on kriging variance, as detailed in section 3.4.1. We have supplemented the uncertainty and the quality flag of VIS_AOD in the gridded product.

*3. There is considerable interannual variability of AOD, if possible could you please consider other AOD products to support the analysis in this study, e.g. MERRA-2 product.*

- **Response:** Thank you for your suggestion. We have added a comparison with MERRA-2 AOD in section 3.3.1.

---

## Author Response (AR4)

**Visibility-derived aerosol optical depth over global land from 1980 to 2021**

**Response to Referee #4.**

***Comments:***

*The authors have addressed most comments raised by previous reviewers.*

*The authors also added confidence intervals to the gridded dataset based on Kriging Variance in responds to previous reviewers' comments on accuracy of the extrapolation in data sparse regions.*

*I have several issues with this approach:*

*comment 1. Kriging is a smooth estimator, which means that it presents a smooth spatial interpolation between observations/ station estimates, whereas AOD is often impacted by small scale phenomena, which are not represented in the observational dataset for data sparse region. This means that Kriging is likely to be a not ideal method for creating a gridded data product and that CIs based on Kriging variance, will therefore underestimate the TRUE uncertainty.*

*comment 2. Therefore, adding Kriging variance-based CIs to the dataset that do not include a very important source of variation/ uncertainty have the potential to give naive data users false confidence in the gridded product. To illustrate this issue, there are large deviations in meridional AOD between the gridded product and MODIS at -120 W and 0E (Figure 11, note the label error in the figure), but Figure 10 indicates very high confidence in the result (green, CI ~ 0.02?). This clear inconsistency indicates to me a problem with the approach.*

**Response to comment 1 and comment 2:**

- Thank you for your suggestion. We have removed the gridded data product in this study, and made corresponding modifications based on station data product in Section 3.4 and Section 3.5. And We also have checked and modified the content of the manuscript.

---

## Author Response (AR5)

Dear Editors,

We would like to submit the revised manuscript entitled "Visibility-derived aerosol optical depth over global land from 1959 to 2021" (ref No.: ESSD-2023-447).

*We have fixed the technical issues and checked the manuscript carefully.*

We deeply appreciate your consideration of our manuscript, and we look forward to receiving evaluation.

Yours sincerely,

Kaicun Wang, on behalf of authors